# Dustin: Draft-Augmented Sparse Verification for Efficient Long-Context Generation with Speculative Decoding

WenHung Lee [* 1]  Jian-Jia Chen [* 1]  Xiaolin Lin [2]  Pei-Shuo Wang [1]  Chi-Chih Chang [3]  Chun-Che Yang [1]
Raymond W. Wang [4]  Ryan Wang [4]  Ning-Chi Huang [1]  Grace Li Zhang [2]  Kai-Chiang Wu [1]

## Abstract

While speculative decoding improves inference throughput for multi-batch long-context Large Language Models (LLMs), its efficiency is often limited by a verification bottleneck where Key-Value (KV) cache loading dominates latency. Existing compression methods fail in this regime: static eviction incurs accuracy loss due to saliency shift, while dynamic selection introduces prohibitive computational overhead during the verification path. We propose Dustin, a sparse verification framework designed for long-context speculative decoding. Dustin integrates lookahead signals from the draft model with historical attention from the target model to identify critical tokens with high fidelity across multi-step verification windows. To reduce recomputation latency, this approach further employs a sparse estimation scheme that restricts importance scoring to a minimal subset of attention heads. Evaluations on PG-19 and LongBench with Qwen2.5-72B demonstrate that Dustin achieves a 27.85× speedup in self-attention and a 9.17× end-to-end decoding speedup at a 32k sequence length, all with negligible accuracy degradation.

## 1. Introduction

Large language models (LLMs) (Achiam et al., 2023; Yang et al., 2024; AI@Meta, 2024) address the rising demand for long-context tasks but face severe memory-bandwidth bottlenecks during auto-regressive decoding (Yuan et al., 2024; Pope et al., 2023). As context length grows, the linear expansion of Key-Value (KV) caches increases the memory

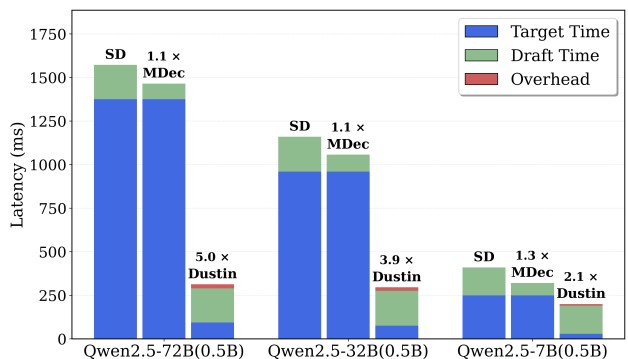

*Figure 1.* **Latency breakdown of a single speculative decoding step.** Experiments are measured with a 32k input length and batch size 16. We compare classic Speculative Decoding (`SD`), MagicDec (`MDec`) (Sadhukhan et al., 2024), and our proposed **Dustin**. The x-axis notation `Target(Draft)` simply indicates the specific target and draft model pair used.

footprint to hundreds of gigabytes, making memory access the primary factor in latency (Kwon et al., 2023; Dao, 2023).

Recent studies (Sun et al., 2024; Sadhukhan et al., 2024; Yang et al., 2025) indicate that speculative decoding effectively improves throughput in multi-batch long-context settings because the computation cost in verification is lower than the substantial overhead of loading the full KV cache. However, the challenge of increasing KV cache loading costs persists. As illustrated in Fig. 1, verification accounts for up to 87.5% of the decoding latency at a 32k input length and batch size of 16, which limits the potential for end-to-end acceleration.

Integrating KV cache compression strategies offers a promising solution, but standard methods are sub-optimal for speculative decoding. Static eviction methods (Xiao et al., 2023; Zhang et al., 2023) permanently discard the context, leading to loss of precision due to saliency shift (Zhao et al., 2025), a phenomenon in which token sets with high attention change over time. Conversely, dynamic selection methods retain the full KV cache but must re-score token importance at every step, adding computation on the verification path. The cost hinges on how importance is estimated: a naïve estimator that materializes attention scores over all heads and layers is prohibitively expensive—our analysis in Sec. 5.4.1 shows

[1]National Yang Ming Chiao Tung University [2]Technische Universität Darmstadt [3]Cornell University [4]Independent Researcher. Correspondence to: WenHung Lee <ja.cs13@nycu.edu.tw>, Jian-Jia Chen <chenjiaj.cs13@nycu.edu.tw>.

*Proceedings of the $43^{rd}$ International Conference on Machine Learning*, Seoul, South Korea. PMLR 306, 2026. Copyright 2026 by the author(s).

its overhead exceeds the latency of full-cache self-attention beyond 4k tokens. Page-level schemes such as Quest (Tang et al., 2024) avoid exact $QK^\top$ via cheap min/max key statistics, yet still incur non-negligible scoring cost and, as we show in Sec. 5.4.1, remain slower than our sparse estimator. Furthermore, we found that solely relying on historical attention scores often results in substantial accuracy drops, as the verification stage processes multiple future steps concurrently.

To address these challenges, we introduce Dustin, a sparse verification approach for multi-batch long-context speculative decoding. By fusing lookahead signals from the draft model with historical attention from the target model, Dustin identifies critical tokens with negligible accuracy loss. To minimize latency, we use a sparse estimation scheme that limits importance scoring to a subset of attention heads, enabling high-speed verification without compromising generation quality.

We evaluate Dustin on LongBench (Bai et al., 2024) and PG-19 (Rae et al., 2019) for the Llama3 (AI@Meta, 2024) and Qwen2.5 (Yang et al., 2024) families, achieving up to a $9.17\times$ decoding speedup on Qwen2.5-72B at a 32k context length with minimal accuracy loss.

In summary, our key contributions are as follows:

- We analyzed the limitations of predicting important tokens solely based on either historical attention scores or lookahead attention scores from the draft model.

- We designed a hybrid token selection policy and combined it with a search algorithm to find a minimal set of attention heads in order to reduce overhead while preserving accuracy.

- Our approach, Dustin, accelerates self-attention computation by $27.85\times$, resulting in a $9.17\times$ decoding phase speedup on PG-19 benchmarks at a 32k input length and batch size 16 on Qwen2.5-72B.

## 2. Background and Related Work

### 2.1. Speculative Decoding

Speculative decoding (SD) (Leviathan et al., 2023; Chen et al., 2023; 2024; Miao et al., 2024) accelerates autoregressive generation by adopting a faster drafter to draft multiple tokens that are then verified in parallel by the target model. Recent work revisits SD for long-context and multi-batch regimes, where inference becomes increasingly memory-traffic dominated. TriForce (Sun et al., 2024) improves scalability via hierarchical speculation. MagicDec (Sadhukhan et al., 2024) shows SD can still yield gains in long-context, large-batch settings by amortizing full-KV target verification over multiple drafted tokens and using sparse-KV draft-

ing to reduce the KV-cache bottleneck. QuantSpec (Tiwari et al., 2025) reduces drafting overhead with quantized weights/KV for self-speculation, while LongSpec (Yang et al., 2025) designs a long-context-oriented drafter with a constant-sized KV cache, together with position-index and attention-aggregation mechanisms for efficient long-context speculative decoding.

### 2.2. KV Cache Eviction

Long-context inference is often bottlenecked by KV cache memory and attention cost, motivating methods that reduce computation by retaining or accessing only a subset of cached tokens. A common line of work performs *attention-guided eviction*, using recent attention weights as a proxy for token importance to decide which KV entries to be kept (Xiao et al., 2023; Liu et al., 2023; Zhang et al., 2023; Li et al., 2024; Cai et al., 2024; Oren et al., 2024; Lin et al., 2025). However, the token set with high attention changes during decoding. SmallKV (Zhao et al., 2025) highlights this as the "saliency shift issue" and mitigates the problem with the help of a small model.

In contrast to irreversible eviction, Quest (Tang et al., 2024) retains the full KV cache and performs *query-aware* selection at each decoding step. It scores KV pages using inexpensive min/max key statistics (avoiding exact $QK^\top$), and attends only to the top-ranked pages, thereby enabling sparse attention without permanent removal.

### 2.3. Target-Side KV Cache Compression in Speculative Decoding

Complementary to compressing the drafter-side state, another line of work reduces *target-side verification* cost by verifying draft tokens using only a sparse subset of the target KV cache. SpecAttn (Shah, 2025) instantiates this idea by using a small draft model to estimate token importance and enabling the target model to verify with token-level sparse KV access, thereby reducing verification-time attention cost in long contexts. This direction is most relevant to our focus on accelerating the *target-model verification* phase under long-context inference.

## 3. Observation

While speculative decoding (SD) accelerates inference for long sequences in large batches, the challenge of the increasing KV cache loading cost persists. Consequently, KV cache compression remains essential. Previous research indicates that permanent KV cache eviction can result in significant information loss due to the **saliency shift issue** (Zhao et al., 2025), caused by dynamic changes in token importance during decoding.

The verification phase involves processing multiple draft

tokens simultaneously. This section evaluates two primary sources to predict and select the most significant $k$ tokens for the sampling range $[i, i + w - 1]$, considering future $w$ tokens:

1. **Historical Attention Scores** extracted from preceding forward passes of the target model.

2. **Lookahead Attention Scores** generated by the draft model during the speculation of future tokens.

We show the limitations of predicting important tokens solely based on historical or lookahead attention scores, and propose a hybrid approach that leverages both strengths to preserve generation quality.

### 3.1. Evaluation Framework: Attention Recovery Rate

We measure how well a KV-selection policy $\pi$ preserves attention using **Attention Recovery Rate (ARR)**. At decoding step $i$, let $\mathcal{V}_i$ be the valid KV cache positions and $A_{i,j}$ represent the normalized attention weight on $j \in \mathcal{V}_i$ such that $\sum_{j \in \mathcal{V}_i} A_{i,j} = 1$. Given a subset $K_i^\pi \subseteq \mathcal{V}_i$ selected by the policy, the ARR is defined as:

$$\mathrm{ARR}_i(\pi) \triangleq \sum_{j \in K_i^\pi} A_{i,j}. \tag{1}$$

**Windowed ARR.** To align with the SD verification of a block of tokens, we calculate the average ARR over a forward window of length $w$:

$$\mathrm{ARR}_i^{(w)}(\pi) \triangleq \frac{1}{w} \sum_{s=0}^{w-1} \mathrm{ARR}_{i+s}(\pi). \tag{2}$$

**SD oracle (reference upper bound).** The optimal subset for maximizing $\mathrm{ARR}_i^{(w)}$ requires access to future attention weights that are unavailable during selection. Therefore, an oracle policy serves as a theoretical upper bound by selecting $K_i$ with full access to future attention:

$$\bar{A}_{i,j}^{(w)} \triangleq \frac{1}{w} \sum_{s=0}^{w-1} A_{i+s, j}, \qquad K_i^{\pi_{\mathrm{orc}}} \triangleq \mathrm{TopK}_k\!\left(\bar{A}_{i,\cdot}^{(w)}\right),$$
$$\tag{3}$$

which yields

$$\mathrm{ARR}_i^{(w)}(\pi_{\mathrm{orc}}) = \sum_{j \in K_i^{\pi_{\mathrm{orc}}}} \bar{A}_{i,j}^{(w)}. \tag{4}$$

Intuitively, ARR is the fraction of the original attention mass preserved by the selected KV tokens: an ARR of 1 means that the selected positions cover all attention mass, while a lower ARR indicates that more attention is lost due to

compression. Therefore, ARR provides a direct measure of how much attention information is retained under a fixed KV cache budget. We further show in Appendix B that ARR is strongly negatively correlated with output-logit KL divergence, suggesting that ARR is a meaningful proxy for sparse-forward fidelity.

Experiments utilize the **Qwen2.5-Instruct** series on the **LongReward** dataset (Zhang et al., 2025) which contains an average context of 13.5k tokens. **Qwen2.5-32B** serves as the primary target model for temporal and layer-wise analysis, while the **0.5B** variant acts as the draft model. For clarity in the following plots, the "KV Cache Budget" refers to the fixed capacity $k$ allocated for the compressed context.

### 3.2. Temporal Decay of Target Historical Signals

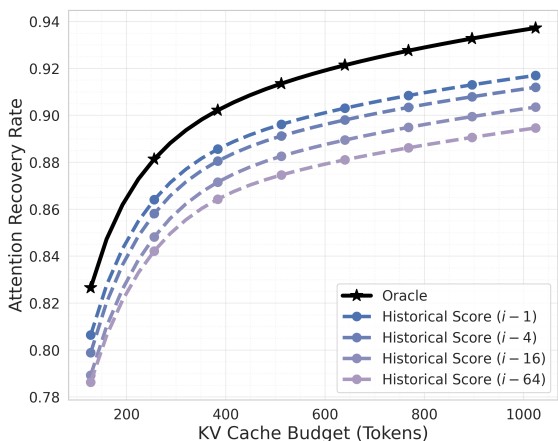

*Figure 2.* Attention recovery rate analysis (Historical Score). Comparison of attention recovery rates using the future average attention (Oracle) versus historical attention scores on the Qwen2.5-32B model. The minimal gap indicates high temporal stability.

This section investigates the validity of **Historical Attention Scores** as a low-overhead proxy for future token importance. Specifically, the analysis evaluates whether token saliency remains consistent across the $w = 4$ verification steps and considers a history-based policy utilizing the past attention of the target model. Context tokens are ranked by $A_{i-\delta,\cdot}$, where $\delta$ denotes the look-back distance in decoding steps. The top-$k$ tokens are retained and compared against the SD Oracle (Eq. 3).

As illustrated in Fig. 2, the results indicate strong short-term consistency. With a KV cache budget of $k = 512$, utilizing the most recent historical attention ($\delta = 1$) achieves an ARR within $1.04\%$ of the Oracle. This suggests that $A_{i-1,\cdot}$ serves as a robust proxy for near-future relevance. However, the verification phase requires processing multiple draft tokens simultaneously, leading to accuracy degradation as draft depth increases. The limits of purely historical strategies are further quantified in the ablation study in Section 5.4.2.

## 3.3. Inconsistency of Draft Lookahead Signals

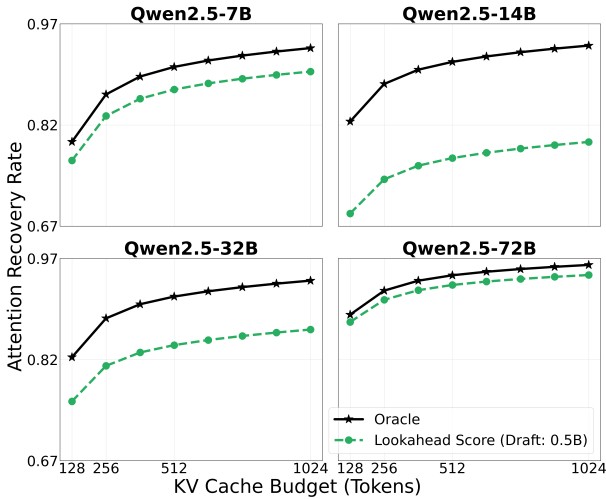

*Figure 3.* Attention recovery rate analysis (Lookahead Scores). Comparison of attention recovery rates on target models (7B-72B) using their own future attention scores (Oracle) versus lookahead scores predicted by a 0.5B draft model. The significant gap in 14B/32B reveals the inconsistency of cross-model prediction.

To mitigate the decay inherent in historical signals, an alternative approach utilizes **lookahead attention scores**. These scores are attention distributions computed directly by the draft model during speculation and serve as real-time indicators of immediate saliency shifts.

While prior methodologies (Zhao et al., 2025; Shah, 2025) rely exclusively on lookahead attention scores from the draft model, this study demonstrates that such policies are prone to inconsistency and risk significant ARR degradation. To evaluate this limitation, a lightweight draft model (0.5B) ranks KV entries via lookahead scores. This policy is measured against the SD Oracle (Eq. 3) to quantify the discrepancy between predicted and ground-truth attention.

As shown in Fig. 3, the ARR exhibits significant variation across different target and draft model pairs. Although the 7B and 72B models align relatively well with the 0.5B draft model, the intermediate 14B and 32B variants suffer from substantial information loss. These findings suggest that selection policies relying solely on signals from draft models lack reliability across diverse model scales. We provide additional evidence across more draft scales and model families in Appendix J.

## 3.4. Hybrid Selection: Integrating Historical and Lookahead Signals

The results in Sec. 3.3 indicate that draft-lookahead importance is unreliable as a standalone global signal. However, these signals can enhance the ARR when utilized as an augmentation to the historical signals of the target model. Fig. 4

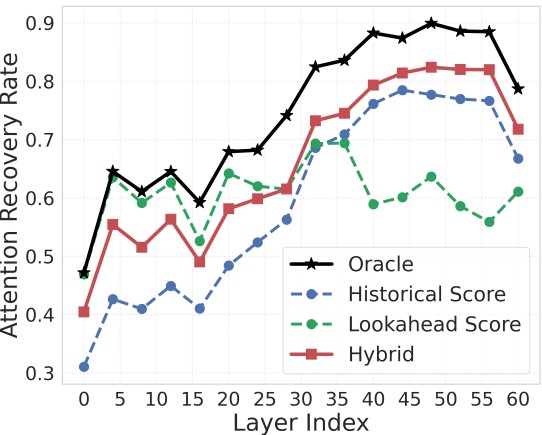

*Figure 4.* Layer-wise attention recovery analysis. Comparison of target-history, draft-lookahead, and hybrid strategies on Qwen2.5-32B. The results highlight structural complementarity: target-historical signals dominate in deeper layers, while draft-lookahead signals excel in early layers (via lookahead).

provides a layer-wise analysis to identify where each signal is most informative, motivating a hybrid construction that leverages their synergistic strengths.

To ensure performance in long-context scenarios, attention scores are computed using only the designated **Semantic Retrieval Heads (SRHs)** across specific layers. As detailed in Sec. 4.2, this selective approach filters noise and aligns with retention quality while significantly reducing computational overhead.

The layer-wise comparative analysis reveals a clear divergence in policy performance:

- **Draft-based Lookahead:** Achieves high ARR in the initial layers but exhibits performance degradation as network depth increases.

- **Target-based History:** Demonstrates an inverse trend, maintaining robustness in deeper layers while performing sub-optimally in the early stages.

This fundamental trade-off between the foresight of the draft model and the reliability of the target model history motivates the proposed hybrid selection strategy. By integrating both attention sources, the hybrid approach maximizes the ARR across the entire architecture. This configuration follows the optimal search parameters identified in Sec. 4.2, ensuring that the distinct strengths of both signals are utilized to maintain high fidelity in the KV cache.

## 4. Methodology

To reduce verification overhead during Speculative Decoding (SD), we introduce Dustin, a sparse verification ap-

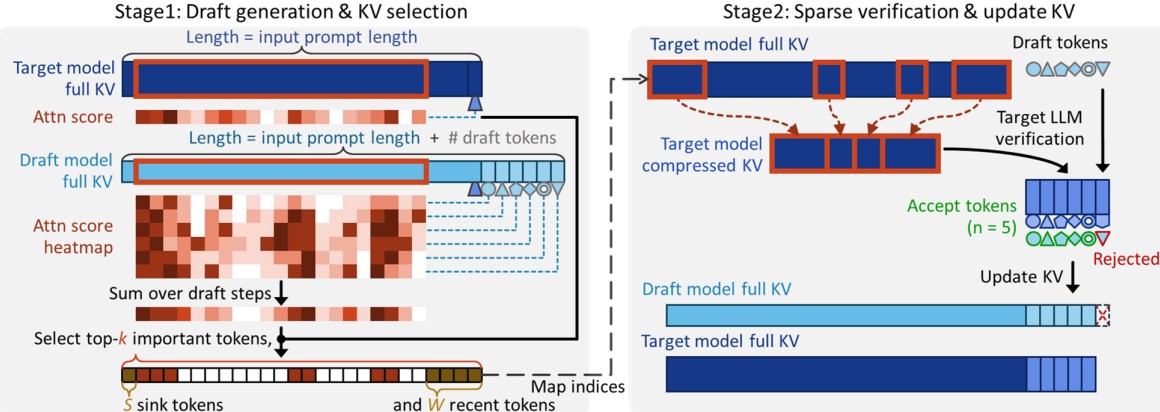

*Figure 5.* Overview of our sparse verification approach. The process begins with hybrid attention aggregation (Eq. 5) to compute a global importance map, followed by Top-K selection (Eq. 6) to determine the final verification set $\mathcal{I}_{verify}$.

proach for efficient Large Language Model (LLM) inference. Dustin identifies critical Key-Value (KV) pairs by integrating target-historical and draft-lookahead attention signals. This approach stems from observations in Sec. 3.4 indicating that combining signals from both models maximizes the Attention Recovery Rate (ARR).

This approach consists of two core components: (1) **hybrid attention aggregation**, which computes a global relevance map for token selection, and (2) **efficient estimation via Semantic Retrieval Heads (SRHs)**, which reduces computational costs by utilizing a small subset of attention heads.

### 4.1. Sparse Verification via Hybrid Attention Aggregation

Fig. 5 illustrates the overall sparse verification workflow in Dustin. During draft generation, Dustin collects attention signals from both the target model and the draft model to estimate the relevance of historical context tokens. These signals are then aggregated into a global importance map, from which Dustin selects a fixed-size verification set. During verification, only the selected KV entries are retrieved to form a compressed target KV cache, which is used to verify the speculative draft tokens. After verification, the accepted tokens are appended to the target KV cache, while rejected draft tokens are discarded.

Dustin determines token relevance by aggregating attention from both the last target token and the speculative draft tokens, providing a more comprehensive signal than the heuristic proxies used in prior works like Quest (Tang et al., 2024). In addition, unlike methods such as SpecAttn (Shah, 2025), which estimate token relevance separately for each layer, Dustin uses a global index set across all layers. This layer-invariant approach simplifies KV retrieval and reduces extra computation overhead.

Formally, let $\mathcal{A}^{\text{Draft}} \in \mathbb{R}^{H_d \times L_d \times N \times \Gamma}$ denote the draft atten-

tion tensor across $H_d$ heads, $L_d$ layers, $N$ context tokens, and $\Gamma$ draft tokens, and let $\mathcal{A}^{\text{Target}} \in \mathbb{R}^{H_t \times L_t \times N \times 1}$ denote the target attention tensor of the last generated token. We compute the global importance vectors by summing attention scores across heads and layers, and also across the $\Gamma$ draft tokens for the draft model, yielding $S_{\text{Draft}} \in \mathbb{R}^N$ and $S_{\text{Target}} \in \mathbb{R}^N$:

$$
\begin{aligned}
S_{\text{Draft}} &= \sum_{h=1}^{H_d} \sum_{l=1}^{L_d} \sum_{\gamma=1}^{\Gamma} \mathcal{A}^{\text{Draft}}_{h,l,n,\gamma}, \\
S_{\text{Target}} &= \sum_{h=1}^{H_t} \sum_{l=1}^{L_t} \sum_{\gamma=1}^{1} \mathcal{A}^{\text{Target}}_{h,l,n,\gamma}.
\end{aligned}
\tag{5}
$$

As shown in Fig. 5, the final verification set $\mathcal{I}_{\text{verify}}$ is constructed through a multi-stage selection process under a fixed budget $k$. Dustin first protects a small prefix of attention sinks ($\mathcal{I}_{\text{sink}}$) and a local window of recent tokens ($\mathcal{I}_{\text{window}}$). It then fills the remaining budget by selecting the top-$m$ tokens according to the draft signal $S_{\text{Draft}}$, followed by the most relevant remaining tokens according to the target signal $S_{\text{Target}}$. This tiered allocation ensures that the most critical historical and lookahead information is preserved for verification:

$$
\begin{aligned}
\mathcal{I}_{\text{prot}} &= \mathcal{I}_{\text{sink}} \cup \mathcal{I}_{\text{window}}, \\
\mathcal{I}_{\text{draft}} &= \text{TopK}(S_{\text{Draft}} \setminus \mathcal{I}_{\text{prot}}, m), \\
\mathcal{I}_{\text{target}} &= \text{TopK}(S_{\text{Target}} \setminus (\mathcal{I}_{\text{prot}} \cup \mathcal{I}_{\text{draft}}), k - m - |\mathcal{I}_{\text{prot}}|), \\
\mathcal{I}_{\text{verify}} &= \mathcal{I}_{\text{prot}} \cup \mathcal{I}_{\text{draft}} \cup \mathcal{I}_{\text{target}}.
\end{aligned}
\tag{6}
$$

### 4.2. Efficient Estimation via Semantic Retrieval Heads

Full reconstruction of attention tensors for both models imposes heavy computational demands, specifically $\mathcal{O}(H_d \cdot$

$L_d \cdot N \cdot \Gamma$) for the draft model and $\mathcal{O}(H_t \cdot L_t \cdot N \cdot 1)$ for the target model. We minimize this overhead by adopting Semantic Retrieval Heads (SRHs) (Lin et al., 2025) as an efficient estimator. We identify a small subset of attention heads in each layer that capture the most significant semantic dependencies, allowing us to estimate token relevance without calculating the full attention map.

### 4.2.1. SEMANTIC RETRIEVAL HEAD SCORING

Fig. 6 illustrates our SRH selection pipeline. Following CompressKV (Lin et al., 2025), we identify SRHs using a layer-wise selection strategy. Specifically, based on profiling scores, only a small subset of heads is retained in each layer to provide their attention scores for KV selection.

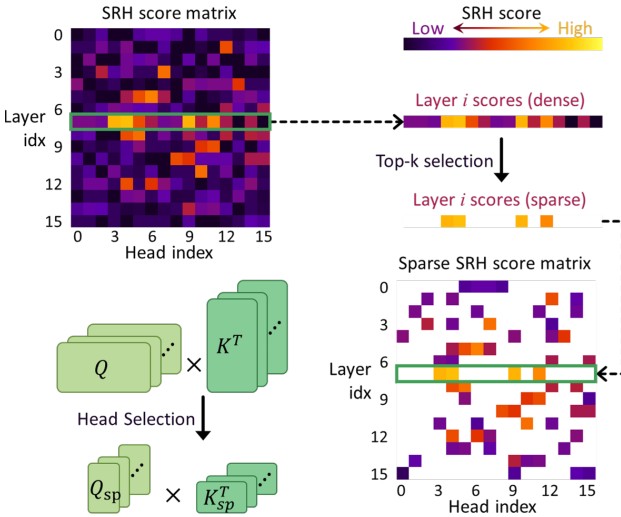

*Figure 6.* Selection process of SRHs. We leverage offline profiling to isolate heads that capture semantic dependencies, enabling efficient online estimation of token relevance using only a fraction of the total attention heads.

### 4.2.2. SPARSE ESTIMATION CONFIGURATION SEARCH

To determine the optimal layer subset and budget split between models, we utilize a multi-stage optimization approach, summarized in Algorithm 1. Crucially, to ensure alignment with the findings in Sec. 3.4, the search utilizes validation set $\Omega$ with **teacher-forced trajectories**. Instead of generic accuracy metrics, we employ the attention recovery rate (ARR) defined in Eq. 2 as the primary optimization objective. These metrics are computed exclusively over the identified SRHs to filter noise. The process decomposes the search space into three phases:

(1) **Greedy target layer search:** We first identify the single target layer $l_t^*$ that maximizes ARR via grid search, establishing a robust cornerstone for online estimation.

(2) **Greedy draft layer search:** Fixing $l_t^*$, we iteratively

---

**Algorithm 1** Sparse Verification Configuration Search

**Require:** Models $\mathcal{M}^{\text{Target}}, \mathcal{M}^{\text{Draft}}$
**Require:** Scores $\mathcal{A}^{\text{Target}}, \mathcal{A}^{\text{Draft}}$ from validation set $\Omega$
**Ensure:** Selected layers $\mathcal{L}_{sel}$, optimal draft split $m$
1: **Phase 1: Greedy Target Layer Search**
2: $l_t^* \leftarrow \arg\max_{l \in \mathcal{L}_t} \text{ARR}(\text{TargetHistorical}(l))$
3: $\mathcal{L}_{sel} \leftarrow \{l_t^*\}$
4: **Phase 2: Greedy Draft Layer Search**
5: **for** $r = 1$ **to** $N_{Draft}$ **do**
6: $\quad J_{step}(d) \triangleq \text{ARR}\left(\text{Hybrid}(\mathcal{L}_{sel} \cup \{d\}), B_{search}\right)$
7: $\quad d^* \leftarrow \arg\max_{d \in \mathcal{L}_d \setminus \mathcal{L}_{sel}} J_{step}(d)$
8: $\quad \mathcal{L}_{sel} \leftarrow \mathcal{L}_{sel} \cup \{d^*\}$
9: **end for**
10: **Phase 3: Budget Optimization (Bayesian)**
11: Let $J_{final}(m) \triangleq \text{ARR}(\text{Hybrid}(\mathcal{L}_{sel}), m)$
12: $m \leftarrow \text{OptunaOptimize}(J_{final}(m))$

---

append draft layers that increase ARR. To maintain computational efficiency during this combinatorial search, we employ a few fixed heuristic budgets $B_{search}$ (e.g., 128, 192, 256 tokens) instead of full grid search.

(3) **Bayesian budget tuning:** Finally, Optuna (Akiba et al., 2019) is employed for Bayesian optimization on the budget parameter $m$, fine-tuning the trade-off between draft-lookahead and target-historical attention scores.

Algorithm 1 is run only once per target–draft model pair after SRH identification, and the resulting configuration is reused across tasks. Appendix I shows that this Dustin-specific offline cost is only 11.9–35.4 minutes for the evaluated 7B–72B model pairs.

### 4.2.3. ONLINE SPARSE ESTIMATION AND COMPLEXITY ANALYSIS

During inference, Dustin recomputes attention scores only for a small subset of the identified Semantic Retrieval Heads (SRHs). This recomputation is necessary for two reasons. First, optimized attention kernels such as FlashAttention (Dao, 2023) avoid writing attention matrices to global memory, making attention weights inaccessible from the normal forward path. Second, Dustin's target verification forward is itself sparse: it attends only to the selected KV entries in $I_{\text{verify}}$. As a result, this forward pass only computes attention scores for the selected KV subset, while scores for unselected context tokens are never produced. Since the next verification step requires ranking tokens over the full candidate context, these sparse-forward attention scores cannot be directly reused for online importance estimation.

By restricting this recomputation to a sparse subset of heads and layers, Dustin minimizes overhead while acquiring the global signals required for token selection. For instance,

in a Qwen2.5-72B/0.5B setup, restricting the estimator to minimal SRHs reduces the theoretical computational cost to approximately $0.8\%$ relative to the full hybrid attention-score calculation. We provide detailed empirical analysis of latency and overhead in Sec. 5.4.1 and Appendix A.

## 5. Experiment

In this section, we evaluate Dustin under long-context and multi-batch inference along two axes: (i) generation quality, measured by long-context task accuracy, and (ii) efficiency, measured by a self-attention latency breakdown and decode-stage throughput (tokens/s). We further conduct ablations to quantify the overhead of online token importance estimation and the impact of budget tuning.

### 5.1. Setup

**Models** We evaluate Dustin on two model families: Llama3 (AI@Meta, 2024) and Qwen2.5 (Yang et al., 2024), each paired with a lightweight same-family draft model. In the main text, we report efficiency results on Qwen2.5-72B and accuracy results on Qwen2.5-72B and Llama-3.3-70B.

**Tasks and Benchmarks** We use PG-19 (Rae et al., 2019) to measure long-context generation throughput and latency, and LongBench (Bai et al., 2024) to evaluate long-context task accuracy across question answering, summarization, few-shot learning, and code-related tasks.

**Implementation Details** All experiments run on NVIDIA H200 GPUs with bfloat16 precision. For the main-text efficiency experiments, Qwen2.5-72B and Llama-3.3-70B are deployed with pipeline parallelism across 4 H200 GPUs. Detailed experimental configurations are provided in Appendix C.

**Accuracy Baselines** Under constrained KV-cache budgets, we compare against: (i) **Vanilla / Lossless SD** methods that preserve the target-model computation without KV compression (including lossless speculative decoding variants such as **MagicDec**); (ii) **StreamingLLM**, a streaming-retention baseline applied to the target model that keeps only sink tokens and a sliding window of recent tokens; (iii) **SnapKV**, a training-free KV-cache compression method that uses attention patterns observed near the end of the prompt to select important prompt KV positions for each attention head, and then keeps the compressed prompt KV cache fixed during decode; and (iv) **Quest**, a block/page-level KV selection method applied to the target model.

**Efficiency Baselines** All efficiency baselines use FlashAttention v2 to ensure a fair comparison. We compare against (i) **Vanilla**, autoregressive decoding without speculative

decoding; (ii) **ClassicSD**, speculative decoding with a same-family draft model and a full KV cache; and (iii) **MagicDec**, speculative decoding with a sparse-KV draft model. To isolate the effect of the verification strategy, all speculative-decoding methods use the same target–draft model pair, sequential draft blocks, and speculative depth for each target model. Detailed configurations, including draft model size and draft length, are provided in Appendix C.

### 5.2. Accuracy Evaluation

We benchmark the accuracy of the sparse verification process against multiple KV cache compression baselines on LongBench under restricted budgets. As detailed in Table 1, Dustin maintains near-lossless performance across most categories and consistently outperforms existing methods. These results indicate that the hybrid selection strategy of this approach effectively preserves critical information, ensuring high fidelity during the verification phase.

### 5.3. Efficiency Evaluation

#### 5.3.1. SELF-ATTENTION LATENCY BREAKDOWN

Fig. 7 breaks down the **self-attention cost in Dustin's target-model verification** into *online importance estimation* and *sparse verification attention*. With a fixed KV budget of 512 tokens, the speedup scales with verification workload: $9.35\times$ (16K, batch 8), and $27.85\times$ (32K, batch 16). More detailed results are provided in Appendix G.

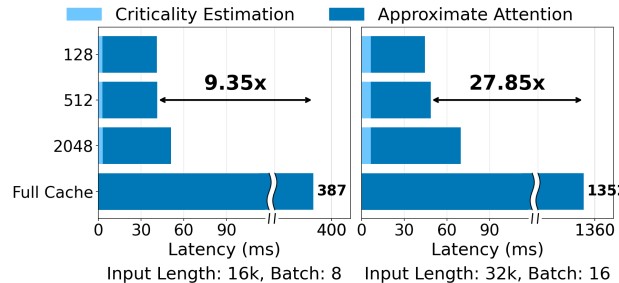

*Figure 7.* Self-attention speedup evaluation on Qwen2.5-72B.

#### 5.3.2. END-TO-END DECODE-STAGE THROUGHPUT

To evaluate decoding efficiency, we sample passages from PG-19 and construct a long-context story continuation task with a fixed instruction prompt. Table 2 reports results for batch sizes of 8 and 16 and context lengths from 8K to 32K tokens. In this evaluation, throughput during the decoding stage (tokens/s) is denoted as *tput*, speedup relative to the Vanilla baseline as $\alpha$, and the average number of accepted tokens per step as $\tau$. Detailed performance metrics for additional experimental configurations are provided in Appendix H.

Dustin attains the highest throughput, and the advantage

*Table 1.* Accuracy evaluation on LongBench under two strict KV cache budgets (512 and 128 tokens). **Bold** indicates the best result, and underline indicates the second-best result among compressed methods under the same target model and KV budget.

| Target Model Compression Method | KV Budget | Single-doc QA | Multi-doc QA | Summarization | Few-shot | Synthetic | Code | Avg. |
|---|---|---|---|---|---|---|---|---|
| *Qwen2.5-72B-Instruct* | | | | | | | | |
| Vanilla / Lossless SD | — | 44.23 | 57.00 | 27.54 | 72.55 | 60.00 | 66.70 | 55.81 |
| StreamingLLM | | 27.61 | 33.82 | 21.70 | 54.57 | 39.84 | 35.02 | 35.25 |
| SnapKV | 512 | 42.80 | **58.77** | 23.02 | 71.21 | 55.00 | 57.29 | 51.90 |
| Quest* | | 36.91 | 41.80 | 25.92 | 64.40 | 55.08 | 57.24 | 47.69 |
| Dustin | | **44.23** | 56.35 | **27.10** | **71.23** | **60.00** | **65.97** | **55.23** |
| StreamingLLM | | 23.27 | 32.49 | 17.58 | 48.48 | 42.69 | 33.68 | 32.72 |
| SnapKV | 128 | 36.57 | **56.89** | 19.80 | 63.71 | 51.63 | 51.54 | 47.08 |
| Quest* | | 23.12 | 25.21 | 21.67 | 52.81 | 49.70 | 46.37 | 37.01 |
| Dustin | | **44.59** | 55.31 | **26.14** | **68.35** | **52.13** | **61.30** | **52.41** |
| *Llama-3.3-70B-Instruct* | | | | | | | | |
| Vanilla / Lossless SD | — | 45.48 | 58.88 | 28.58 | 70.75 | 53.63 | 49.64 | 50.94 |
| StreamingLLM | | 32.36 | 39.35 | 23.01 | 51.55 | 51.05 | 41.98 | 39.61 |
| SnapKV | 512 | 44.45 | **59.58** | 24.87 | 68.75 | 53.39 | 58.82 | 52.41 |
| Quest* | | 40.34 | 52.24 | 27.18 | 66.63 | **54.61** | 51.95 | 48.96 |
| Dustin | | **45.27** | 59.10 | **27.76** | **70.14** | 53.40 | **59.15** | **53.23** |
| StreamingLLM | | 27.13 | 39.78 | 19.02 | 46.43 | 52.44 | 39.49 | 36.94 |
| SnapKV | 128 | 39.17 | 58.02 | 22.07 | 64.55 | **54.63** | 58.53 | 50.32 |
| Quest* | | 23.66 | 27.93 | 21.09 | 47.94 | 44.20 | 35.67 | 33.21 |
| Dustin | | **45.55** | **59.02** | **26.75** | **68.18** | 53.96 | **62.78** | **53.81** |

The symbol * denotes that Quest does not apply KV compression to decoder layers 0–1, which slightly exceeds KV budget.

grows with both batch size and context length. On **Qwen2.5-72B**, for batch size 8, speedup increases from $3.01\times$ (8K) to $6.61\times$ (32K). For batch size 16, speedup increases from $4.76\times$ (8K) to $9.17\times$ (32K). The same trend holds on **Llama-3.3-70B**, where Dustin reaches 6.28× (batch 8) and 7.18× (batch 16) at 32K, consistently outperforming ClassicSD and MagicDec.

## 5.4. Ablation Study

To quantify the contribution of each attention source, we evaluate three Dustin variants. **Dustin-H** denotes the full hybrid design, which combines *target-based historical* attention with *draft-based lookahead* attention to identify important context tokens. This variant uses 4 selected heads from the target model and 12 selected heads from the draft model. **Dustin-T** disables the *draft-based lookahead* component and performs token selection using only *target-based historical* attention, with 4 selected target heads. **Dustin-D** disables the *target-based historical* component and performs token selection using only *draft-based lookahead* attention, with 12 selected draft heads.

### 5.4.1. ANALYSIS OF IMPORTANCE ESTIMATION OVERHEAD

We evaluate online importance estimation latency across varying input lengths using Qwen2.5-72B (target) and Qwen2.5-0.5B (draft), with a batch size of 8 and a KV budget of 512. Fig. 8 reports normalized latency relative to

a full-cache forward pass, denoted as **flash-attn**. All values are normalized by the full-cache self-attention latency. Quest-16 denotes Quest with page size 16. Naïve materializes attention scores from all heads in both the target and draft models. Dustin-H introduces only a negligible overhead compared to flash-attn, while being much faster than Quest-16 across all context lengths. In contrast, Naïve materializes attention from all heads and layers and becomes cost-prohibitive, exceeding full-cache self-attention beyond 4K tokens, underscoring the necessity of attention head selection for efficient online estimation. Across the three head-selection variants (Dustin-H, Dustin-T, Dustin-D), the estimator remains lightweight, with Dustin-H staying near ~1% of flash-attn at 16K–32K. Compared with Dustin-T and Dustin-D, Dustin incurs only a small additional cost to combine both signals; later results show this marginal overhead yields higher accuracy than either single-source variant.

### 5.4.2. HISTORICAL AND LOOKAHEAD POLICY COMPARISON

Table 3 compares the performance of our hybrid strategy against single-source baselines. By integrating complementary signals from both target-historical and draft-lookahead attention, Dustin-H consistently obtains relatively good accuracy across diverse benchmarks. In contrast, single-source variants (Dustin-T and Dustin-D) exhibit performance fluctuations depending on the task nature. This result demon-

*Table 2.* Efficiency evaluation across different context lengths and batch sizes on Qwen2.5-72B-Instruct and Llama-3.3-70B-Instruct

| Method | Batch Size | Context Length | | | | | | | | | | | |
| --- | --- | --- | --- | --- | --- | --- | --- | --- | --- | --- | --- | --- | --- |
| | | 8K | | | 16K | | | 24K | | | 32K | | |
| | | tput | $\tau$ | $\alpha$ | tput | $\tau$ | $\alpha$ | tput | $\tau$ | $\alpha$ | tput | $\tau$ | $\alpha$ |
| *Qwen2.5-72B-Instruct* | | | | | | | | | | | | | |
| Vanilla | 8 | 58.06 | - | 1.00× | 38.75 | - | 1.00× | 29.05 | - | 1.00× | 23.26 | - | 1.00× |
| ClassicSD | | 101.82 | 2.2 | 1.75× | 67.66 | 2.2 | 1.75× | 50.47 | 2.2 | 1.74× | 40.83 | 2.2 | 1.76× |
| MagicDec | | 98.90 | 2.1 | 1.70× | 65.67 | 2.0 | 1.69× | 49.91 | 2.1 | 1.72× | 39.63 | 2.0 | 1.70× |
| Dustin | | 174.86 | 2.2 | **3.01×** | 172.37 | 2.3 | **4.45×** | 162.12 | 2.2 | **5.58×** | 153.67 | 2.2 | **6.61×** |
| Vanilla | 16 | 76.23 | - | 1.00× | 46.03 | - | 1.00× | 32.95 | - | 1.00× | 25.70 | - | 1.00× |
| ClassicSD | | 145.40 | 2.3 | 1.91× | 89.26 | 2.2 | 1.94× | 63.00 | 2.2 | 1.91× | 48.68 | 2.1 | 1.89× |
| MagicDec | | 143.67 | 2.1 | 1.88× | 89.27 | 2.1 | 1.94× | 64.63 | 2.1 | 1.96× | 50.38 | 2.0 | 1.96× |
| Dustin | | 362.50 | 2.2 | **4.76×** | 324.86 | 2.2 | **7.06×** | 276.34 | 2.2 | **8.39×** | 235.81 | 2.2 | **9.17×** |
| *Llama-3.3-70B-Instruct* | | | | | | | | | | | | | |
| Vanilla | 8 | 59.01 | - | 1.00× | 39.44 | - | 1.00× | 29.52 | - | 1.00× | 23.61 | - | 1.00× |
| ClassicSD | | 115.91 | 2.7 | 1.96× | 75.31 | 2.6 | 1.91× | 55.14 | 2.8 | 1.87× | 43.41 | 2.9 | 1.84× |
| MagicDec | | 122.80 | 2.7 | 2.08× | 79.28 | 2.5 | 2.01× | 60.67 | 2.7 | 2.06× | 48.48 | 2.8 | 2.05× |
| Dustin | | 237.72 | 2.9 | **4.03×** | 205.66 | 2.7 | **5.21×** | 175.43 | 2.9 | **5.94×** | 148.20 | 2.8 | **6.28×** |
| Vanilla | 16 | 77.38 | - | 1.00× | 46.69 | - | 1.00× | 33.45 | - | 1.00× | 26.00 | - | 1.00× |
| ClassicSD | | 151.33 | 2.7 | 1.96× | 87.68 | 2.6 | 1.88× | 61.73 | 2.8 | 1.85× | 47.42 | 2.9 | 1.82× |
| MagicDec | | 162.47 | 2.7 | 2.10× | 98.79 | 2.6 | 2.12× | 71.20 | 2.8 | 2.13× | 55.47 | 2.8 | 2.13× |
| Dustin | | 407.96 | 2.9 | **5.27×** | 291.87 | 2.8 | **6.25×** | 226.85 | 2.8 | **6.78×** | 186.69 | 2.9 | **7.18×** |

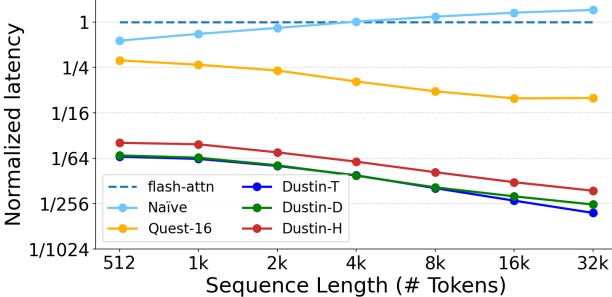

*Figure 8.* Normalized Latency of Online Importance Estimation

*Table 3.* Accuracy comparison of three Dustin variants (Dustin-T, Dustin-D, and Dustin-H) on TriviaQA and MultiNews benchmarks.

| Method | Tri. QA | | M.News | |
| --- | --- | --- | --- | --- |
| | Acc. | $\Delta$ | Acc. | $\Delta$ |
| Dustin-T | 77.58% | -6.41% | 23.95% | -1.12% |
| Dustin-D | 80.51% | -3.48% | 24.84% | -0.23% |
| Dustin-H | 83.63% | -0.36% | 25.19% | +0.12% |

strates that the hybrid approach provides a stable mechanism for importance estimation, ensuring robust performance preservation where isolated signals might fail. Results on additional datasets are provided in Appendix K.

## 6. Conclusion

We presented **Dustin**, a sparse verification framework that addresses the KV-cache loading bottleneck limiting speculative decoding in long-context, multi-batch regimes. Motivated by the observation that neither target-historical nor draft-lookahead attention is individually reliable across model scales and verification depths, Dustin fuses both signals to identify critical tokens with high fidelity, while restricting importance scoring to a small set of Semantic Retrieval Heads to keep online estimation overhead near 0.8% of a full hybrid computation. Evaluations on PG-19 and

LongBench across the Llama3 and Qwen2.5 families show that Dustin delivers up to a 27.85× self-attention speedup and a 9.17× end-to-end decoding speedup on Qwen2.5-72B at a 32k context length, with near-lossless accuracy that consistently surpasses streaming and page-level baselines, and an advantage that widens as batch size and context grow. Two limitations point to future work: the budget-allocation parameter is fixed via offline profiling, so an input-aware dynamic scheme could further close the gap to lossless verification; and because Dustin indexes the full history rather than evicting tokens, it does not reduce the KV-cache memory footprint, and therefore—unlike eviction methods—does not by itself allow longer sequences to fit within a fixed GPU memory budget. We hope Dustin offers a practical foundation for high-throughput long-context inference.

## Impact Statement

This paper presents work whose goal is to advance the field of machine learning. There are many potential societal consequences of our work, none of which we feel must be specifically highlighted here.

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

## A. Detailed online sparse estimation and complexity analysis

After identifying SRHs via offline profiling, we efficiently estimate token importance by aggregating attention scores solely from this small subset on-the-fly. In practice, only the attention scores required by this subset are materialized and accumulated during inference, while all remaining heads and layers are processed normally.

This sparsity is critical for maintaining high inference speed. Since efficient attention kernels like FlashAttention utilize online softmax to avoid materializing the full attention matrix to global memory, explicitly storing token estimation scores from *all* layers and heads would induce significant I/O overhead. Our method preserves the throughput advantages of these optimized kernels for the majority of the computation by restricting score materialization to selected SRHs.

To quantify this benefit, consider a Speculative Decoding (SD) configuration with Qwen2.5-72B as the target model ($H_t = 64, L_t = 80$) and Qwen2.5-0.5B as the draft model ($H_d = 14, L_d = 24$) speculating four tokens ($\Gamma = 4$). By downsizing the estimator to 1 target layer with 4 heads and 3 draft layers with 4 heads (denoted as $L^*, H^*$), the overhead ratio relative to the full hybrid calculation is derived as follows:

$$
\begin{aligned}
\text{Overhead Ratio} &= \frac{(H_d^* \cdot L_d^* \cdot \Gamma) + (H_t^* \cdot L_t^* \cdot 1)}{(H_d \cdot L_d \cdot \Gamma) + (H_t \cdot L_t \cdot 1)} \\
&= \frac{(4 \cdot 3 \cdot 4) + (4 \cdot 1 \cdot 1)}{(14 \cdot 24 \cdot 4) + (64 \cdot 80 \cdot 1)} \\
&= \frac{48 + 4}{1344 + 5120} \approx 0.8\%
\end{aligned}
\tag{7}
$$

This minimal computational footprint ensures that the sparse verification process introduces negligible latency.

Furthermore, it is important to distinguish our optimization strategy from methods that target the sequence length dimension ($N$). While our work reduces computational overhead by sparsifying the model architecture dimensions—specifically the number of heads ($H$) and layers ($L$)—other recent works, such as Quest (Tang et al., 2024), focus on compressing the context dimension during the importance estimation phase. Specifically, Quest aggregates token-level Key statistics (e.g., minimum and maximum values) into page-level metadata. This allows it to estimate the relevance of a block of tokens using a single computation, effectively reducing the sequence length dimension for estimation from $N$ to $N/\text{page size}$. Our Semantic Retrieval Head (SRH) method is orthogonal to such dimension-reduction techniques. Whereas Quest compresses the effective sequence length for estimation, we minimize the number of attention mechanisms required. Consequently, our approach focuses on lightweight architectural profiling rather than dynamic context compression, though both directions could potentially be combined for further efficiency.

## B. Relationship Between ARR and Output Distribution Distortion

To further validate whether ARR reflects the fidelity of sparse KV selection, we conduct an additional controlled experiment that measures the relationship between ARR and output distribution distortion. Instead of evaluating a specific KV-selection policy, we sample KV subsets with different retained attention mass and compare the resulting sparse forward pass against a full-cache reference. This removes the bias introduced by any particular selection algorithm and isolates the intrinsic relationship between retained attention mass and output distribution distortion.

For each model and input sample, we first run a full-cache forward pass and greedily generate a fixed sequence of tokens. During this reference run, we store the full-cache logits and the corresponding attention distribution at each decoding step. We then replay the same generated token sequence under sparse KV caches with sampled KV subsets spanning different ARR ranges. For each sparse forward pass, we compute two metrics: ARR, which measures the fraction of full-cache attention mass retained by the selected KV tokens, and the KL divergence between the next-token distributions induced by the sparse-cache and full-cache logits.

Formally, let $p_i^{\text{full}}$ and $p_i^{\text{sparse}}$ denote the next-token probability distributions produced by the full-cache and sparse-cache forward passes at decoding step $i$, respectively. We measure output distribution distortion using

$$
D_{\text{KL}}\left(p_i^{\text{full}} \,\|\, p_i^{\text{sparse}}\right).
\tag{8}
$$

For each sampled KV subset, we average ARR and KL divergence across decoding steps, and then compute the Pearson correlation between the averaged ARR and averaged KL divergence.

We run this experiment on a GovReport sample using both **Llama-3.1-8B-Instruct** and **Qwen2.5-7B-Instruct**. The KV budget is fixed to 512 tokens, with 4 sink tokens and 16 recent tokens always preserved. The resulting ARR–KL correlations are shown in Table 4. ARR exhibits a consistent negative correlation with KL divergence across both models, indicating that higher ARR generally corresponds to lower output distribution distortion.

*Table 4.* Correlation between ARR and output-logit KL divergence. Higher ARR is associated with lower output distribution distortion.

| Model | ARR–KL Correlation |
|---|---|
| Llama-3.1-8B-Instruct | -0.9490 |
| Qwen2.5-7B-Instruct | -0.8079 |

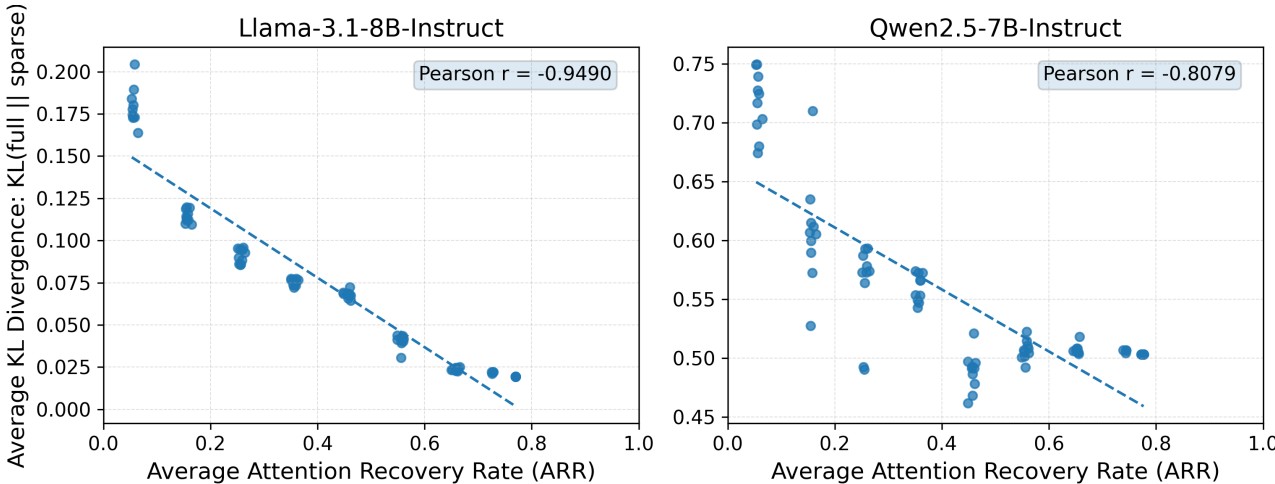

*Figure 9.* Relationship between Attention Recovery Rate (ARR) and output-logit KL divergence. The left and right panels show results for **Llama-3.1-8B-Instruct** and **Qwen2.5-7B-Instruct**, respectively. Each point corresponds to one sampled KV subset, with ARR and KL divergence averaged across decoding steps. The negative trend indicates that higher ARR generally leads to lower output distribution distortion.

These results support ARR as a meaningful proxy for sparse-forward fidelity. Although ARR is computed purely from attention mass, it is predictive of output distribution distortion: preserving more attention mass leads to logits that are closer to the full-cache reference. Fig. 9 visualizes this relationship, while Table 4 reports the corresponding Pearson correlations.

## C. Detailed Experimental Setup

### C.1. Models

**Extending Qwen2.5 Context Length to 64K via YaRN** All Qwen2.5 models are configured to support up to 64K context length by enabling YaRN-based RoPE scaling, following the official recommendation for length extrapolation beyond the default 32,768-token setting. Concretely, YaRN is activated through the model configuration (e.g., a RoPE scaling factor of 2 with the original maximum position embeddings set to 32,768 and scaling type set to `yarn`), and all Qwen2.5 experiments that require long contexts use this setup.

### C.2. Implementation Details

**FlashAttention-v2 MagicDec for Fair Efficiency Comparison** To ensure a fair and controlled efficiency comparison, a FlashAttention v2 version of MagicDec is implemented in-house. This removes kernel-level confounders and aligns the attention backend across efficiency baselines, so the measured throughput and latency differences primarily reflect algorithmic design (e.g., speculation and KV handling) rather than disparate attention implementations.

**Hardware Setup and Pipeline Parallelism**    All efficiency-evaluation experiments run on NVIDIA H200 GPUs with bfloat16 precision and pipeline parallelism. Model deployment follows a size-dependent GPU allocation: (i) **4×H200** for **Qwen2.5-72B**, **Qwen2.5-32B**, and **Llama-3.3-70B**; (ii) **2×H200** for **Qwen2.5-14B**, **Qwen2.5-7B**, and **Llama-3.1-8B**. This hardware policy is applied consistently across the corresponding efficiency baselines.

**Dustin KV Selection Configuration**    For Dustin, the protected token set consists of 4 attention sink tokens and a recent window of 16 tokens. These protected tokens are always retained and are counted as part of the total KV budget $k$. The remaining budget is allocated to tokens selected by the hybrid draft-lookahead and target-historical attention signals following Eq. 6.

## C.3. Accuracy baselines setup

**StreamingLLM Configuration**    For baselines utilizing the StreamingLLM strategy, we configure the KV cache to maintain a fixed budget $k$. This policy strictly retains the first 4 tokens as attention sinks and the most recent $k - 4$ tokens as a sliding local window.

**Quest Configuration for Accuracy Evaluation**    For LongBench accuracy evaluation, Quest is configured with a page size of **16** (i.e., block granularity of 16 tokens) for its block-level selection. Following the official Quest practice, *layers 0–1 do not apply KV compression/selection*; block-level selection is only enabled for higher layers.

## C.4. Efficiency baselines setup

**Speculative Decoding Configuration.**    For all speculative-decoding efficiency baselines, we pair each target model with a lightweight draft model from the same model family. For a given target model, the same draft model is used across Classic SD, MagicDec, and Dustin, so that the comparison isolates the effect of the verification strategy rather than differences in drafting quality or draft-model cost.

All methods use standard sequential draft blocks rather than tree-based speculation. At each speculative step, the draft model generates a block of $\Gamma$ consecutive draft tokens, which are then verified in parallel by the target model. The speculative depth $\Gamma$ is fixed for each target model and shared across Classic SD, MagicDec, and Dustin. The complete per-model draft configuration is summarized in Table 5.

*Table 5.* Speculative decoding configurations used in efficiency experiments.

| Target model | Draft model | Draft structure | Depth $\Gamma$ |
|---|---|---|---|
| Qwen2.5-72B-Instruct | Qwen2.5-0.5B-Instruct | Sequential | 4 |
| Qwen2.5-32B-Instruct | Qwen2.5-0.5B-Instruct | Sequential | 4 |
| Qwen2.5-14B-Instruct | Qwen2.5-0.5B-Instruct | Sequential | 3 |
| Qwen2.5-7B-Instruct | Qwen2.5-0.5B-Instruct | Sequential | 3 |
| Llama-3.3-70B-Instruct | Llama-3.2-1B-Instruct | Sequential | 4 |
| Llama-3.1-8B-Instruct | Llama-3.2-1B-Instruct | Sequential | 3 |

**Draft KV-cache Configuration for Classic SD and MagicDec.**    Classic SD follows the standard speculative decoding procedure, where the draft model maintains a full KV cache over the entire input context. In contrast, MagicDec reduces draft-stage latency by applying StreamingLLM-style KV retention to the draft model. Specifically, MagicDec keeps the first 4 attention-sink tokens and the most recent $k - 4$ tokens as a sliding local window in the draft-model KV cache. Dustin follows the same draft-model configuration as the corresponding speculative-decoding baseline for each target model, while applying sparse verification on the target-model side.

**Target-side Verification Configuration.**    Classic SD and MagicDec verify draft tokens using the full target-model KV cache and therefore preserve lossless target-side verification. Dustin instead performs target-side sparse verification using the selected verification set $\mathcal{I}_{\text{verify}}$ described in Sec. 4.1. Unless otherwise specified, the target-side KV budget is set to $k = 512$ in efficiency experiments. This setting is shared across Dustin's sparse verification experiments and is independent of the speculative depth $\Gamma$.

# D. Additional Accuracy Results on Llama3 and Qwen2.5 Model Series

Table 6 and Table 7 report detailed LongBench accuracy results on Llama-3.1-8B-Instruct and additional Qwen2.5 models (32B, 14B, and 7B) under multiple constrained KV-cache budgets (512/256/128). These experiments complement the main-text evaluation on Llama-3.3-70B-Instruct and Qwen2.5-72B-Instruct models and assess whether the effectiveness of Dustin generalizes across model scales and model families.

Across all sizes and budgets, Dustin consistently maintains accuracy close to the FullKV oracle while substantially outperforming streaming baselines and block-level selection (Quest). The advantage becomes more pronounced as the KV budget tightens, where competing methods suffer noticeable degradation, whereas Dustin preserves most task performance.

This trend is stable across diverse task categories, including single- and multi-document QA, summarization, few-shot learning, synthetic retrieval, and code generation, indicating that the proposed hybrid criticality estimation remains effective under both capacity-constrained and scale-varying regimes.

Overall, the results indicate that the benefits of Dustin are not limited to a specific model size or model family, but transfer reliably to Llama-3.1-8B-Instruct and across Qwen2.5 models (7B–32B), confirming the robustness and scalability of the proposed verification strategy.

*Table 6.* Performance comparison of Dustin with StreamingLLM, Quest and FullKV on LongBench for Llama3 series models (Llama-3.1-8B-Instruct)

| Target Model Compression Method | KV Budget | Single-doc QA | Multi-doc QA | Summarization | Few-shot | Synthetic | Code | Avg. |
|---|---|---|---|---|---|---|---|---|
| *Llama-3.1-8B-Instruct* | | | | | | | | |
| ClassicSD / MagicDec | — | 42.34 | 44.15 | 29.11 | 69.38 | 52.75 | 59.72 | 50.58 |
| StreamingLLM | | 28.35 | 30.11 | 23.16 | 50.51 | 47.11 | 38.31 | 36.00 |
| Quest | 512 | 33.60 | 34.70 | 27.21 | 55.94 | 51.33 | 54.92 | 43.90 |
| Dustin | | 42.26 | 44.26 | 28.63 | 69.08 | 52.75 | 59.68 | 50.45 |
| StreamingLLM | | 25.84 | 29.73 | 21.16 | 47.60 | 47.06 | 36.83 | 34.39 |
| Quest | 256 | 27.61 | 24.64 | 24.45 | 48.67 | 48.37 | 49.86 | 38.15 |
| Dustin | | 42.15 | 44.31 | 27.72 | 68.77 | 52.59 | 58.11 | 49.81 |
| StreamingLLM | | 25.02 | 29.46 | 19.25 | 43.94 | 45.23 | 35.79 | 32.86 |
| Quest | 128 | 18.53 | 18.32 | 19.96 | 34.19 | 41.43 | 40.54 | 29.54 |
| Dustin | | 41.95 | 43.99 | 25.03 | 68.42 | 52.84 | 57.15 | 49.02 |

*Table 7.* Performance comparison of Dustin with StreamingLLM, Quest and FullKV on LongBench for Qwen2.5 series models (Qwen2.5-32B-Instruct, Qwen2.5-14B-Instruct, Qwen2.5-7B-Instruct)

| Target Model Compression Method | KV Budget | Single-doc QA | Multi-doc QA | Summarization | Few-shot | Synthetic | Code | Avg. |
|---|---|---|---|---|---|---|---|---|
| *Qwen2.5-32B-Instruct* | | | | | | | | |
| ClassicSD / MagicDec | — | 42.43 | 54.53 | 27.37 | 67.57 | 56.00 | 42.78 | 47.52 |
| StreamingLLM | | 24.32 | 27.79 | 20.90 | 45.75 | 12.50 | 27.47 | 27.33 |
| Quest | 512 | 31.30 | 36.88 | 24.16 | 59.55 | 54.97 | 40.00 | 40.41 |
| Dustin | | 41.37 | 54.50 | 26.60 | 66.75 | 56.25 | 41.73 | 46.85 |
| StreamingLLM | | 20.87 | 27.13 | 18.57 | 42.51 | 13.00 | 26.25 | 25.56 |
| Quest | 256 | 23.58 | 31.55 | 21.00 | 52.73 | 50.64 | 39.54 | 36.25 |
| Dustin | | 41.29 | 54.10 | 25.71 | 64.87 | 53.50 | 40.53 | 45.72 |
| StreamingLLM | | 19.94 | 26.36 | 15.93 | 40.35 | 13.25 | 25.89 | 24.46 |
| Quest | 128 | 18.49 | 22.36 | 16.87 | 43.38 | 46.22 | 34.15 | 29.97 |
| Dustin | | 41.12 | 53.70 | 24.76 | 62.81 | 37.92 | 39.77 | 43.28 |
| *Qwen2.5-14B-Instruct* | | | | | | | | |
| ClassicSD / MagicDec | — | 42.16 | 52.95 | 27.36 | 71.64 | 55.88 | 59.29 | 52.27 |
| StreamingLLM | | 23.79 | 28.36 | 20.25 | 47.98 | 5.36 | 32.38 | 28.15 |
| Quest | 512 | 29.27 | 30.51 | 25.04 | 60.66 | 49.73 | 43.67 | 39.83 |
| Dustin | | 41.11 | 52.36 | 26.68 | 70.88 | 55.70 | 58.15 | 51.46 |
| StreamingLLM | | 20.91 | 26.27 | 18.24 | 44.45 | 8.75 | 32.12 | 26.80 |
| Quest | 256 | 21.64 | 23.02 | 23.27 | 53.37 | 43.88 | 38.24 | 34.00 |
| Dustin | | 40.21 | 51.19 | 26.18 | 69.88 | 54.42 | 56.43 | 50.31 |
| StreamingLLM | | 19.68 | 26.06 | 16.00 | 41.45 | 13.50 | 30.36 | 25.78 |
| Quest | 128 | 15.41 | 14.34 | 20.47 | 44.22 | 33.08 | 35.57 | 27.92 |
| Dustin | | 40.59 | 49.75 | 25.57 | 69.49 | 38.86 | 53.72 | 47.59 |
| *Qwen2.5-7B-Instruct* | | | | | | | | |
| ClassicSD / MagicDec | — | 40.54 | 44.28 | 27.76 | 69.92 | 53.50 | 64.11 | 51.46 |
| StreamingLLM | | 23.03 | 24.21 | 21.98 | 45.21 | 9.50 | 34.56 | 28.23 |
| Quest | 512 | 23.61 | 21.71 | 24.83 | 52.61 | 40.15 | 47.93 | 36.39 |
| Dustin | | 40.21 | 43.83 | 27.45 | 68.75 | 53.50 | 61.45 | 50.40 |
| StreamingLLM | | 19.05 | 22.98 | 19.47 | 42.89 | 11.50 | 32.74 | 26.40 |
| Quest | 256 | 19.97 | 15.93 | 22.55 | 45.05 | 34.24 | 42.26 | 31.21 |
| Dustin | | 40.41 | 43.17 | 27.38 | 67.89 | 49.25 | 60.42 | 49.44 |
| StreamingLLM | | 19.32 | 23.14 | 17.19 | 41.03 | 12.25 | 31.44 | 25.54 |
| Quest | 128 | 14.54 | 10.29 | 19.35 | 34.61 | 26.02 | 35.79 | 24.73 |
| Dustin | | 39.87 | 42.75 | 26.76 | 67.84 | 35.50 | 55.04 | 46.29 |

## E. Additional Accuracy Results Compared with SmallKV

Given the difference in base models, we focus our comparison on the accuracy drop relative to the respective FullKV baselines rather than absolute scores. As shown in Table 8, our method outperforms SmallKV in four out of five benchmarks, demonstrating significantly smaller performance degradation. Notably, in Single-doc QA, our method achieves near-lossless performance with a negligible drop of 0.33, whereas SmallKV suffers a 5.9-point decline. Similar trends are observed in Multi-doc QA, Summarization, and Few-shot tasks, where our method consistently maintains higher fidelity to the original baseline. The only exception is the Code benchmark, where SmallKV retains better performance.

*Table 8.* Performance comparison of Dustin with SmallKV and FullKV on LongBench for Qwen2 series models (Qwen2-7B, Qwen2.5-7B-Instruct)

| Target Compression Method | KV Budget | Single-doc QA | Multi-doc QA | Summarization | Few-shot | Code |
|---|---|---|---|---|---|---|
| | | *Qwen2-7B* | | | | |
| Vanilla / Lossless SD | — | 39.04 | 40.78 | 22.31 | 70.07 | 56.95 |
| SmallKV | $\approx 512$ | 33.14(-5.9) | 37.54(-3.24) | 21.55(-0.76) | 68.39(-1.68) | 56.83(**-0.12**) |
| | | *Qwen2.5-7B-Instruct* | | | | |
| Vanilla / SD Methods | — | 40.54 | 44.28 | 27.76 | 69.92 | 64.11 |
| Dustin | 512 | 40.21(**-0.33**) | 43.83(**-0.45**) | 27.45(**-0.31**) | 68.75(**-1.17**) | 61.45(-2.66) |

## F. Additional Efficiency Evaluation Compared with SpecAttn

We further compare Dustin with SpecAttn, a sparse verification method that selects KV tokens using attention scores computed during speculative decoding. Since SpecAttn does not provide an official open-source implementation, we implement a SpecAttn-style baseline based on our Dustin codebase. Specifically, we modify Dustin's selection procedure to compute token importance using attention scores from every layer and every head of the draft model, following the per-layer and per-head relevance estimation strategy of SpecAttn, instead of using Dustin's hybrid global aggregation design.

Table 9 reports the measured speedups on **Qwen2.5-72B-Instruct** with batch size 16. Dustin consistently outperforms SpecAttn across all evaluated context lengths, achieving higher speedups at 8K, 16K, and 32K contexts.

*Table 9.* Efficiency comparison between SpecAttn and Dustin across different context lengths. Results are measured on Qwen2.5-72B-Instruct with batch size 16.

| Method | Batch Size | Context Length | | |
|---|---|---|---|---|
| | | **8K** | **16K** | **32K** |
| SpecAttn | 16 | 3.91$\times$ | 5.62$\times$ | 7.11$\times$ |
| Dustin | 16 | **4.76$\times$** | **7.06$\times$** | **9.17$\times$** |

## G. More Self-Attention Latency Breakdown

Figure 10 presents the latency decomposition across four distinct workload configurations on Qwen2.5-72B. We isolate the **Criticality Estimation Overhead**—which encompasses the time spent on **SRH scoring** (utilizing the configuration determined via **offline layer search**) to identify the optimal verification set—from the **Sparse Verification Attention**, which represents the actual computation of the target model attending to the selected 512 tokens.

**Speedup Analysis** Dustin significantly accelerates the **full cache speculative verification self-attention** baseline. Using a fixed 512-token budget, speedups scale with workload: $9.35\times$ (Batch 8) and $15.58\times$ (Batch 16) at 16k context, rising to $16.57\times$ and $27.85\times$ respectively at 32k. This confirms that sparse verification effectively alleviates memory bandwidth bottlenecks in long-context, large-batch scenarios.

**Criticality Estimation Overhead** The online estimation overhead is minimal, constituting only $0.46\%$–$0.75\%$ of the full cache latency. Even relative to the sparse verification step, the overhead remains moderate ($7.53\%$–$14.88\%$). This

demonstrates that the cost of SRH scoring and layer search is negligible compared to the massive savings in attention computation.

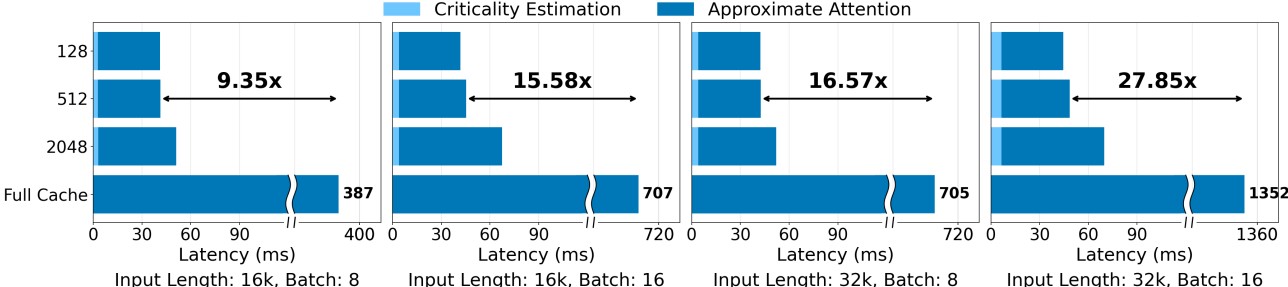

*Figure 10.* Detailed latency breakdown of the target model verification phase on Qwen2.5-72B. The charts compare the latency of **Full Cache** (Baseline) against **Dustin**, decomposing the latter into *Criticality Estimation* (light blue) and *Approximate Attention* (dark blue). Our estimation overhead remains negligible across all settings, while the sparse attention yields massive latency reductions, particularly at longer context lengths (32k) and larger batch sizes (16).

## H. Additional End-to-End Decode-Stage Throughput on the Llama3 and Qwen2.5 Model Series

To extensively validate the scalability of our approach, we report end-to-end decode-stage throughput on the Llama3 and Qwen2.5 model families across various scales (70B, 32B, 14B, 8B, 7B). Table 10 and Table 11 present the results for the Llama3 series and Qwen2.5 series, respectively.

**Performance on Llama3 Series**  As shown in Table 10, Dustin consistently outperforms all baselines on **Llama-3.1-8B-Instruct**, where it achieves up to $3.17\times$ speedup at 32K context with batch size 16, surpassing MagicDec ($1.89\times$). Notably, as the context length increases from 8K to 32K, the throughput of the Vanilla baseline drops significantly (e.g., from 215.30 to 67.42 tokens/s at batch 16), reflecting the severe KV cache bottleneck. In contrast, Dustin maintains significantly higher throughput, demonstrating its ability to mitigate long-context overhead. Compared to ClassicSD and MagicDec, which plateau below $1.9\times$ even at 32K, Dustin's advantage becomes increasingly prominent at longer contexts.

**Performance on Qwen2.5 Series**  Table 11 details the results for **Qwen2.5-32B**, **14B**, and **7B**. Dustin demonstrates strong scalability across all sizes. For **Qwen2.5-32B**, the speedup reaches an impressive $7.81\times$ at 32K context (batch 16), far exceeding the $\sim 2.2\times$ speedup of MagicDec. Even for the smaller **Qwen2.5-7B**, where the compute ratio between the target and draft models is less favorable for speculative decoding, Dustin still achieves a substantial $2.33\times$ speedup at 32K context (batch 16). This contrasts with ClassicSD and MagicDec, which struggle to exceed $1.4\times$ in the same setting.

**Impact of Batch Size on Scalability**  A consistent finding across both tables is the positive correlation between batch size and speedup. Increasing the batch size from 8 to 16 amplifies the relative speedup of Dustin (e.g., on Qwen2.5-32B at 32K, $\alpha$ improves from $5.85\times$ to $7.81\times$). This phenomenon confirms that our sparse verification strategy is particularly effective in bandwidth-constrained regimes (large batch, long context), where reducing the volume of KV cache transfers yields the highest returns.

*Table 10.* Efficiency evaluation on Llama-3.1-8B-Instruct across different context lengths and batch sizes

| Method | Batch Size | Context Length | | | | | | | | | | | |
|---|---|---|---|---|---|---|---|---|---|---|---|---|---|
| | | 8K | | | 16K | | | 24K | | | 32K | | |
| | | tput | $\tau$ | $\alpha$ | tput | $\tau$ | $\alpha$ | tput | $\tau$ | $\alpha$ | tput | $\tau$ | $\alpha$ |
| **Llama-3.1-8B-Instruct** | | | | | | | | | | | | | |
| Vanilla | 8 | 169.70 | - | 1.00× | 110.40 | - | 1.00× | 80.31 | - | 1.00× | 63.13 | - | 1.00× |
| ClassicSD | | 215.11 | 2.6 | 1.27× | 149.50 | 2.7 | 1.35× | 107.71 | 2.8 | 1.34× | 84.80 | 2.6 | 1.34× |
| MagicDec | | 236.27 | 2.5 | 1.39× | 171.80 | 2.6 | 1.56× | 133.37 | 2.7 | 1.66× | 110.05 | 2.5 | 1.74× |
| Dustin | | 313.30 | 2.7 | 1.85× | 264.58 | 2.7 | 2.40× | 214.07 | 2.8 | 2.67× | 180.14 | 2.7 | 2.85× |
| Vanilla | 16 | 215.30 | - | 1.00× | 124.43 | - | 1.00× | 87.21 | - | 1.00× | 67.42 | - | 1.00× |
| ClassicSD | | 289.06 | 2.6 | 1.34× | 168.33 | 2.7 | 1.35× | 117.92 | 2.7 | 1.35× | 91.36 | 2.7 | 1.36× |
| MagicDec | | 337.75 | 2.5 | 1.57× | 217.74 | 2.6 | 1.75× | 160.96 | 2.6 | 1.85× | 127.11 | 2.6 | 1.89× |
| Dustin | | 510.88 | 2.7 | 2.37× | 351.17 | 2.7 | 2.82× | 266.08 | 2.8 | 3.05× | 213.64 | 2.8 | 3.17× |

*Table 11.* Efficiency evaluation on Qwen2.5-32B, 14B, and 7B-Instruct across different context lengths and batch sizes

| Method | Batch Size | Context Length | | | | | | | | | | | |
|---|---|---|---|---|---|---|---|---|---|---|---|---|---|
| | | 8K | | | 16K | | | 24K | | | 32K | | |
| | | tput | $\tau$ | $\alpha$ | tput | $\tau$ | $\alpha$ | tput | $\tau$ | $\alpha$ | tput | $\tau$ | $\alpha$ |
| **Qwen2.5-32B-Instruct** | | | | | | | | | | | | | |
| Vanilla | 8 | 80.62 | - | 1.00× | 52.17 | - | 1.00× | 38.56 | - | 1.00× | 30.49 | - | 1.00× |
| ClassicSD | | 125.09 | 2.3 | 1.55× | 88.45 | 2.3 | 1.70× | 68.15 | 2.3 | 1.77× | 56.04 | 2.3 | 1.84× |
| MagicDec | | 128.09 | 2.2 | 1.59× | 90.54 | 2.1 | 1.74× | 69.52 | 2.2 | 1.80× | 57.61 | 2.2 | 1.89× |
| Dustin | | 205.00 | 2.3 | 2.54× | 200.55 | 2.3 | 3.84× | 186.89 | 2.2 | 4.85× | 178.35 | 2.2 | 5.85× |
| Vanilla | 16 | 102.24 | - | 1.00× | 60.32 | - | 1.00× | 42.73 | - | 1.00× | 33.14 | - | 1.00× |
| ClassicSD | | 192.57 | 2.2 | 1.88× | 122.97 | 2.4 | 2.04× | 85.88 | 2.3 | 2.01× | 68.16 | 2.4 | 2.06× |
| MagicDec | | 193.29 | 2.1 | 1.89× | 127.52 | 2.3 | 2.11× | 92.10 | 2.3 | 2.16× | 74.70 | 2.3 | 2.25× |
| Dustin | | 426.06 | 2.3 | 4.17× | 379.82 | 2.4 | 6.30× | 309.21 | 2.2 | 7.24× | 258.69 | 2.2 | 7.81× |
| **Qwen2.5-14B-Instruct** | | | | | | | | | | | | | |
| Vanilla | 8 | 113.99 | - | 1.00× | 73.24 | - | 1.00× | 53.33 | - | 1.00× | 41.96 | - | 1.00× |
| ClassicSD | | 146.87 | 2.1 | 1.29× | 96.79 | 2.0 | 1.32× | 77.29 | 2.2 | 1.45× | 61.23 | 2.1 | 1.46× |
| MagicDec | | 147.94 | 2.1 | 1.30× | 101.75 | 2.0 | 1.39× | 79.72 | 2.1 | 1.49× | 63.52 | 2.0 | 1.51× |
| Dustin | | 233.19 | 2.2 | 2.05× | 221.47 | 2.1 | 3.02× | 208.28 | 2.1 | 3.91× | 194.36 | 2.1 | 4.63× |
| Vanilla | 16 | 142.80 | - | 1.00× | 82.77 | - | 1.00× | 58.14 | - | 1.00× | 44.87 | - | 1.00× |
| ClassicSD | | 212.73 | 2.2 | 1.49× | 130.65 | 2.0 | 1.58× | 94.75 | 2.1 | 1.63× | 72.65 | 2.1 | 1.62× |
| MagicDec | | 218.78 | 2.1 | 1.53× | 137.91 | 2.0 | 1.67× | 102.09 | 2.1 | 1.76× | 80.34 | 2.1 | 1.79× |
| Dustin | | 443.91 | 2.1 | 3.11× | 397.19 | 2.1 | 4.80× | 330.63 | 2.2 | 5.69× | 278.11 | 2.1 | 6.20× |
| **Qwen2.5-7B-Instruct** | | | | | | | | | | | | | |
| Vanilla | 8 | 255.05 | - | 1.00× | 206.22 | - | 1.00× | 157.37 | - | 1.00× | 126.35 | - | 1.00× |
| ClassicSD | | 244.14 | 2.3 | 0.96× | 194.80 | 2.4 | 0.94× | 158.03 | 2.4 | 1.00× | 132.62 | 2.3 | 1.05× |
| MagicDec | | 247.25 | 2.1 | 0.97× | 202.91 | 2.3 | 0.98× | 171.16 | 2.3 | 1.09× | 145.94 | 2.1 | 1.16× |
| Dustin | | 300.73 | 2.3 | 1.18× | 280.57 | 2.4 | 1.36× | 257.39 | 2.3 | 1.64× | 239.76 | 2.3 | 1.90× |
| Vanilla | 16 | 402.94 | - | 1.00× | 249.36 | - | 1.00× | 180.80 | - | 1.00× | 141.47 | - | 1.00× |
| ClassicSD | | 385.72 | 2.4 | 0.96× | 270.66 | 2.4 | 1.09× | 198.47 | 2.4 | 1.10× | 155.42 | 2.3 | 1.10× |
| MagicDec | | 402.30 | 2.2 | 1.00× | 296.98 | 2.3 | 1.19× | 235.08 | 2.3 | 1.30× | 194.38 | 2.1 | 1.37× |
| Dustin | | 553.79 | 2.3 | 1.37× | 486.50 | 2.3 | 1.95× | 393.90 | 2.3 | 2.18× | 329.61 | 2.2 | 2.33× |

## I. Porting overhead to new models

Dustin requires a small one-time offline cost when porting to a new target–draft model pair. This cost comes from two stages: (i) identifying Semantic Retrieval Heads (SRHs) for the target model, and (ii) collecting full cache attention traces and running the sparse verification configuration search in Algorithm 1. Importantly, this procedure is performed once per model pair and does not require per-task or per-dataset recalibration.

**SRH identification cost.** We follow the SRH identification procedure of CompressKV (Lin et al., 2025), where retrieval-oriented heads are identified once for a model and then reused across downstream tasks. To quantify the practical cost, we re-measured the SRH identification pipeline on Llama-3.1-8B-Instruct using a single A100 GPU. As shown in Table 12, the profiling time remains within tens of minutes even for long calibration contexts. Specifically, the identification takes 13 minutes at 16K input length, 19 minutes at 32K, and 27 minutes at 50K. This indicates that SRH profiling is a lightweight offline preparation step rather than a deployment-time bottleneck.

*Table 12.* Offline SRH identification time on Llama-3.1-8B-Instruct using a single A100 GPU.

| Max Input Length | Identification Time |
|:---:|:---:|
| 16K | 13 min |
| 32K | 19 min |
| 50K | 27 min |

**Dustin configuration search cost.** After SRHs are identified, Dustin performs a one-time model-pair-specific configuration search. This stage consists of a full cache run to collect target-historical and draft-lookahead attention traces, followed by the search procedure described in Algorithm 1. Table 13 reports the measured overhead across several target–draft pairs. On a single H100 GPU, the total cost is 11.9 minutes for Llama-3.1-8B with a 1B draft, 13.7 minutes for Qwen2.5-7B with a 0.5B draft, and 18.1 minutes for Qwen2.5-14B with a 0.5B draft. For larger models evaluated with two H100 GPUs, the total cost is 24.1 minutes for Qwen2.5-32B and 35.4 minutes for Qwen2.5-72B. These results show that even for 70B-scale models, the additional Dustin-specific porting overhead remains below one hour.

*Table 13.* One-time Dustin porting overhead after SRH identification. The overhead consists of full cache attention collection and Algorithm 1 configuration search.

| Target + Draft Model Pair | Hardware | Attention Collection | Search | Total |
|:---|:---:|:---:|:---:|:---:|
| Llama-3.1-8B + 1B | H100 $\times$ 1 | 7.5 min | 4.4 min | 11.9 min |
| Qwen2.5-7B + 0.5B | H100 $\times$ 1 | 6.6 min | 7.1 min | 13.7 min |
| Qwen2.5-14B + 0.5B | H100 $\times$ 1 | 10.6 min | 7.5 min | 18.1 min |
| Qwen2.5-32B + 0.5B | H100 $\times$ 2 | 15.8 min | 8.3 min | 24.1 min |
| Qwen2.5-72B + 0.5B | H100 $\times$ 2 | 26.6 min | 8.8 min | 35.4 min |

**No per-task recalibration.** The above cost is amortized across all future inference workloads using the same target–draft pair. Once the SRH set, selected layers, and draft–target budget split are obtained, Dustin keeps them fixed during deployment. This is consistent with our robustness analysis, where SRH selection remains stable under changes in calibration size, haystack source, and context length. Therefore, porting Dustin to a new model mainly requires a short offline profiling phase, while downstream tasks can directly reuse the same configuration without additional task-specific tuning.

## J. Additional Generalization Analysis of Draft-Lookahead Signals

This appendix provides additional evidence for the observations in Sec. 3.3 and Sec. 3.4. In the main text, we use Qwen2.5-0.5B as the draft model to analyze the inconsistency of draft-lookahead signals across Qwen2.5 target models. Here, we extend the analysis to additional draft scales and model families to better characterize when draft-lookahead attention is reliable and when the hybrid target-history/draft-lookahead strategy is needed.

### J.1. Experimental Setup

We follow the same ARR-based evaluation framework as Sec. 3.1. For each target–draft pair, the draft model generates speculative tokens and provides lookahead attention scores, which are used to rank context KV positions. The selected top-$k$ positions are then evaluated against the SD oracle defined in Eq. 4.

For this appendix analysis, we use 200 evaluation samples from the `zai-org/LongReward-10k` dataset. The average context length is approximately 13.5K tokens, and the maximum output length is set to 512 tokens. We set the verification window length to $w = 4$ and use the same KV-cache budgets as in the main observation experiments.

We evaluate the following target–draft model pairs:

- **Qwen2.5 pairs:** We use Qwen2.5-0.5B and Qwen2.5-1.5B as draft models, each paired with Qwen2.5-7B, Qwen2.5-14B, Qwen2.5-32B, and Qwen2.5-72B as target models.

- **Llama3 pairs:** We use Llama-3.2-1B and Llama-3.2-3B as draft models, each paired with Llama-3.1-8B and Llama-3.3-70B as target models.

The goal of this analysis is not to exhaustively tune every target–draft pair, but to examine whether the non-monotonic behavior observed in Fig. 4 persists across different draft scales and architectures.

### J.2. Cross-Model ARR with Additional Draft Scales

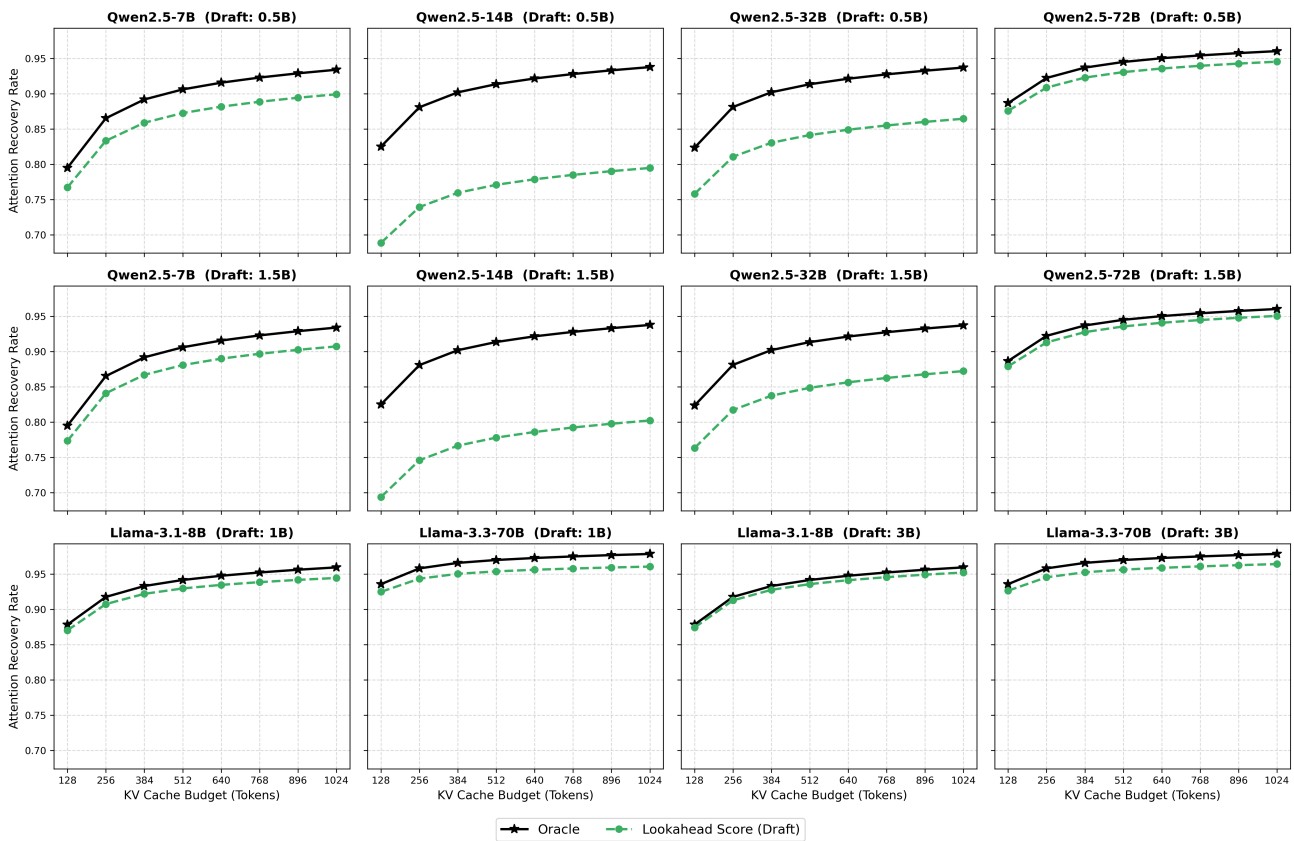

*Figure 11.* Additional cross-model ARR analysis across Qwen2.5 and Llama3 target–draft pairs. We compare oracle ARR with draft-lookahead ARR using Qwen2.5-0.5B, 1.5B and Llama-3.2-1B, 3B as draft models across their corresponding larger target models. The results further characterize the model-pair-dependent reliability of draft-lookahead signals.

Fig. 11 reports the ARR of draft-lookahead selection across all target–draft pairs in our appendix setup, covering both Qwen2.5 and Llama3 families. The results show that draft-lookahead reliability varies significantly across model pairs. In

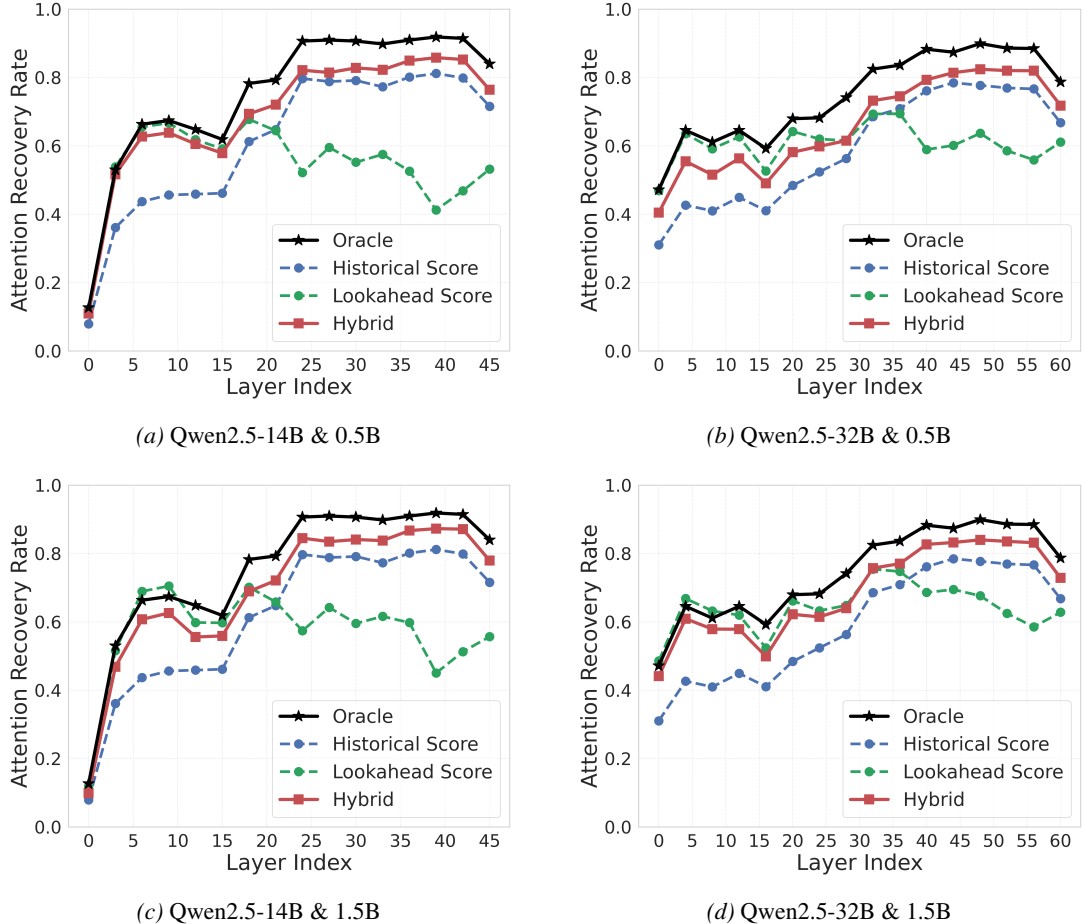

*Figure 12.* Layer-wise comparison of target-history, draft-lookahead, and hybrid selection on high-gap Qwen2.5 target–draft pairs. These results show that draft-lookahead and target-history provide complementary signals across layers, motivating the hybrid construction.

the Qwen2.5 family, Qwen2.5-14B and Qwen2.5-32B still show a clear gap from the oracle even with a larger 1.5B draft, while Qwen2.5-72B aligns much more closely with the draft signal. In contrast, the Llama3 pairs maintain consistently small gaps between draft-lookahead ARR and oracle ARR. These results indicate that draft-lookahead attention can provide useful future-aware signals, but its reliability is not uniform across model families or target–draft pairs. This supports the need for a hybrid strategy that incorporates target-side history when draft-side signals are unreliable.

### J.3. Layer-Wise Complementarity on High-Gap Model Pairs

We next focus on the high-gap Qwen2.5 pairs identified in Fig. 11, namely Qwen2.5-14B and Qwen2.5-32B paired with Qwen2.5-0.5B, 1.5B drafts. For these pairs, we apply the head and layer selection procedure described in Sec. 4.2, and compare target-history, draft-lookahead, and hybrid selection layer by layer.

Fig. 12 shows that draft-lookahead and target-history provide complementary layer-wise signals. The hybrid selection therefore tracks the oracle more closely than either signal alone, supporting Dustin's hybrid construction.

## K. Additional historical and lookahead comparison policy

In this section, we present a fine-grained analysis of the impact of our hybrid selection strategy compared to single-source baselines across all 16 LongBench tasks. We focus on the robustness of the selected budget configuration and the stability of performance across diverse datasets. Furthermore, we investigate the theoretical upper bound of our framework by assuming an optimal dynamic selection of the mixing parameter.

## K.1. Configuration of the Mixing Parameter $m$

As visualized in Fig. 13, the performance of the verification framework is sensitive to the budget allocation parameter $m$, which controls the trade-off between Draft-Lookahead and Target-Historical signals. Specifically, the blue lines represent the **Target-Historical only** policy (where $m = 0$), the green lines represent the **Draft-Lookahead only** policy (where $m$ is maximized), and the orange/red curves illustrate the **Hybrid** performance as $m$ varies.

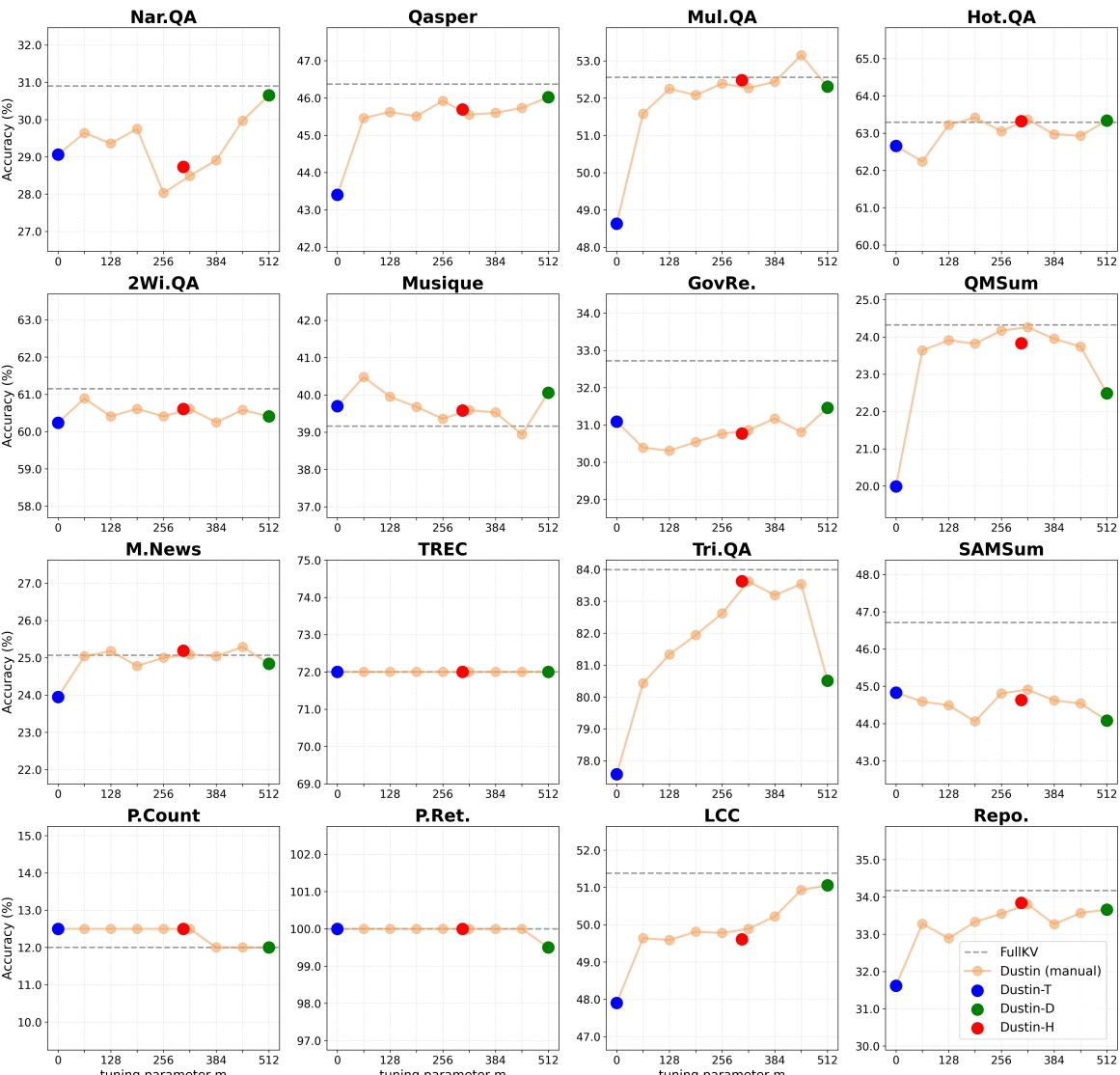

*Figure 13.* Impact of the budget tuning parameter $m$ on accuracy recovery across different benchmarks. Blue indicates Target-Historical only, green indicates Draft-Lookahead only, and the orange/red curve represents the Hybrid approach.

It is important to note that the optimal $m$ used in our main experiments (Dustin-H) was determined via Bayesian optimization using Optuna on the **LongReward** dataset, acting as a proxy for general long-context capability. While this global parameter setting may not correspond to the global maximum for every individual benchmark in LongBench, the consistent performance observed highlights the generalization capability of Dustin, as it functions effectively without requiring dataset-specific hyperparameter tuning for each downstream task.

## K.2. Performance Stability and Robustness Analysis

Table 14 details the accuracy gap ($\Delta$, in %) relative to the FullKV baseline. We compare three variations along with an Oracle baseline:

- **Dustin-T**: Relies solely on Target-Historical attention.

- **Dustin-D**: Relies solely on Draft-Lookahead attention.

- **Dustin-H**: Our proposed Hybrid method using the fixed Optuna-tuned $m$.

- **Oracle (Best)**: A theoretical upper bound that dynamically selects the best policy (T, D, or H) for each specific benchmark.

**Average Performance and Win Rate**  Dustin-H achieves the smallest average accuracy degradation among the fixed policies, with only a $0.59\%$ drop from the lossless baseline. It outperforms Dustin-T and Dustin-D, which incur average accuracy drops of $1.91\%$ and $0.71\%$, respectively. In head-to-head comparisons across the 16 benchmarks, Dustin-H further demonstrates superior versatility:

- **vs. Dustin-T:** Dustin-H wins in **9** tasks, loses in 4, and ties in 3.

- **vs. Dustin-D:** Dustin-H wins in **9** tasks, loses in 6, and ties in 1.

This indicates that combining signals allows the model to adapt to tasks where one signal type might be insufficient (e.g., retrieval tasks favoring draft lookahead vs. reasoning tasks favoring historical context).

**Robustness and Potential of Dynamic Allocation**  A critical advantage of the Hybrid approach is its stability. We calculated the standard deviation of accuracy penalties (defined as the magnitude of negative drops). The results highlight the robustness of our method compared to single-source strategies:

- **Dustin-T**: $\sigma_{\text{drop}} = 1.86\%$

- **Dustin-D**: $\sigma_{\text{drop}} = 1.03\%$

- **Dustin-H**: $\sigma_{\text{drop}} = \mathbf{0.83}\%$

- **Oracle**: $\sigma_{\text{drop}} = 0.52\%$

Comparing Dustin-H to the **Oracle** (Dynamic $m$), which achieves an average drop of just $-0.27\%$ and a standard deviation of $0.52\%$, we observe that our fixed-parameter approach captures a significant portion of the recoverable accuracy. The gap between Dustin-H ($-0.59\%$) and the Oracle ($-0.27\%$) suggests that while Dustin-H is highly effective and safe, future work exploring task-aware dynamic $m$ selection could further close the gap to lossless verification.

*Table 14.* Detailed accuracy comparison ($\Delta$ relative to FullKV, %). Negative values indicate accuracy loss. **Oracle** represents the best result selected from T, D, and H for each task.

| Benchmark | Dustin-T | Dustin-D | Dustin-H | Oracle |
|---|---|---|---|---|
| NarrativeQA | -1.84 | **-0.25** | -2.17 | -0.25 |
| Qasper | -2.97 | **-0.35** | -0.68 | -0.35 |
| MultiFieldQA | -3.93 | -0.25 | **-0.08** | -0.08 |
| HotpotQA | -0.63 | **+0.05** | +0.03 | +0.05 |
| 2WikiMQA | -0.91 | -0.74 | **-0.54** | -0.54 |
| Musique | +0.54 | **+0.90** | +0.42 | +0.90 |
| GovReport | -1.63 | **-1.26** | -1.95 | -1.26 |
| QMSum | -4.33 | -1.83 | **-0.49** | -0.49 |
| MultiNews | -1.12 | -0.23 | **+0.12** | +0.12 |
| TREC | +0.00 | +0.00 | +0.00 | +0.00 |
| TriviaQA | -6.41 | -3.48 | **-0.36** | -0.36 |
| SAMSum | **-1.88** | -2.63 | -2.08 | -1.88 |
| PassageCount | **+0.50** | +0.00 | **+0.50** | +0.50 |
| PassageRet | **+0.00** | -0.50 | **+0.00** | +0.00 |
| LCC | -3.48 | **-0.32** | -1.77 | -0.32 |
| RepoBench | -2.55 | -0.51 | **-0.33** | -0.33 |
| **Average** | -1.91 | -0.71 | **-0.59** | -0.27 |

## L. Limitations

While Dustin effectively reduces verification latency and improves throughput in long-context scenarios, we identify two primary limitations in the current framework that point towards future research directions.

**Static and Dynamic Budget Allocation** As analyzed in the ablation study (Appendix K), our current approach relies on a fixed budget allocation parameter $m$ (determining the ratio between draft-lookahead and target-historical). Although optimizing $m$ via offline profiling yields robust performance across averaged benchmarks, the comparison with the "Oracle" policy reveals that different tasks exhibit distinct preferences for validation signals. The use of a static, globally optimized scalar inevitably results in a performance gap compared to an ideal dynamic strategy. Future work could explore a lightweight, input-aware mechanism to dynamically adjust $m$ on-the-fly, potentially closing the gap to near-lossless verification.

**Computation and Memory Capacity** Dustin is primarily designed as a sparse verification framework to mitigate the computational overhead and memory bandwidth bottleneck during the speculative decoding verification phase. While it significantly reduces the number of active tokens loaded during attention computation, it currently employs a global indexing mechanism that selects from the full history. Unlike permanent token eviction methods which physically remove tokens to reduce the VRAM footprint, Dustin does not inherently expand the maximum supportable context length constrained by GPU memory capacity.

