# OpenReview forum: "Dustin: Draft-Augmented Sparse Verification for Efficient Long-Context Generation with Speculative Decoding"
_ICML.cc/2026/Conference — ICML 2026 regular_

### Official Review · Reviewer_tC4w · 2026-03-06

**Soundness:** 3
**Presentation:** 2
**Significance:** 3
**Originality:** 3
**Overall Recommendation:** 4
**Confidence:** 3

**Summary:**

This paper proposes Dustin, a sparse verification framework for accelerating long-context speculative decoding. The key observation is that in multi-batch long-context settings, KV cache loading during the target model's verification phase dominates latency (up to 87.5%). Dustin addresses this by performing verification on a sparse subset of the KV cache, selected via a hybrid strategy that combines: (1) historical attention scores from the target model's last forward pass, and (2) lookahead attention scores from the draft model's speculative tokens. To minimize the overhead of computing these scores, Dustin identifies a small set of Semantic Retrieval Heads (SRHs) through a three-phase offline search (greedy target layer → greedy draft layer → Bayesian budget tuning).

**Compliance With Llm Reviewing Policy:**

Affirmed.

**Final Justification:**

Based on the paper and especially the rebuttal, I changed my evaluation from 3 (Weak Reject) to 4 (Weak Accept) because the authors substantially clarified the role and limitations of ARR, provided strong ARR–KLD evidence, added tail-sensitive failure analysis and robustness/overhead measurements, and included the missing SpecAttn baseline and an explanation for the Fig. 3 non-monotonicity.

**Key Questions For Authors:**

**Q1.** Can you explain the non-monotonic behavior in Fig. 3—why does the 0.5B draft align well with 7B and 72B but poorly with 14B and 32B? Have you tested with draft models other than 0.5B? *A convincing explanation or broader validation would improve confidence in the hybrid strategy's generalizability and could raise the Soundness rating.*

**Q2.** What is the wall-clock time for the three-phase offline configuration search (Algorithm 1), and how sensitive are the results to the choice and size of the validation set? *If the overhead is modest and generalizes well, this would alleviate concerns about practical deployability.*

**Q3.** Are there observed failure modes (e.g., tasks requiring precise long-range retrieval) where the sparse KV approximation leads to noticeable quality degradation beyond what the averaged LongBench scores reveal? *Per-task failure analysis would help practitioners assess when the method is safe to deploy.*

**Limitations:**

Partially discussed. Appendix H identifies two limitations (static budget allocation and memory capacity), which is appreciated. However, the following are not discussed: (1) the fundamental loss of lossless guarantees when compressing the target's KV cache during verification, (2) the offline search cost and its scalability.

**Strengths And Weaknesses:**

### **Strengths**

**S1. Important and well-scoped problem.** The verification bottleneck in long-context speculative decoding is a real and increasingly relevant problem. The latency breakdown in Fig. 1 clearly motivates the work by showing verification accounts for up to 87.5% of decoding latency.

**S2. Strong empirical results.** The efficiency gains are impressive and scale well with context length and batch size (3.01× at 8K to 9.17× at 32K for batch 16). The accuracy evaluation in Table 1 shows Dustin maintains near-lossless performance (55.23 vs. 55.81 on Qwen2.5-72B at budget=512), substantially outperforming StreamingLLM and Quest. Results span multiple model families (Qwen2.5, Llama-3) and scales (7B–72B).

### **Weaknesses**

**W1. ARR does not guarantee lossless output, and the SD oracle framing is misleading.** Standard speculative decoding guarantees lossless output by verifying against the unmodified target distribution. Dustin compresses the target's KV cache, fundamentally altering this distribution, yet provides no formal analysis linking ARR to output fidelity (e.g., KL divergence between full-KV and sparse-KV distributions). Furthermore, the SD oracle (Eq. 3–4) maximizes ARR with future attention access, but maximizing ARR ≠ minimizing output divergence—so it does not establish whether any given ARR level is sufficient for faithful generation. The theoretical foundation of the entire framework is thus incomplete.

**W2. Fig. 3 shows inconsistent cross-model patterns.** The 0.5B draft aligns well with 7B and 72B targets but poorly with 14B and 32B—a non-monotonic behavior that is unexplained and counter-intuitive. This raises concerns about whether the hybrid strategy generalizes to arbitrary target-draft pairs. The paper acknowledges this but offers no satisfying explanation.

**W3. The three-phase offline search is complex and its overhead is unquantified.** Algorithm 1 requires grid search over target layers, iterative draft layer search, and Bayesian optimization via Optuna—all on a validation set with teacher-forced trajectories, repeated per target-draft pair. The paper does not report wall-clock time, required validation set size, or sensitivity to validation set choice, making practical deployability hard to assess.

**W4. Missing comparison with LongSpec and SpecAttn.** LongSpec (Yang et al., 2025), cited in Related Work, addresses the same problem and reports up to 3.26× speedup. SpecAttn (Shah, 2025) is discussed conceptually but also absent from experiments. These omissions leave the competitive standing unclear.

**W5. Presentation issues.** (1) Fig. 5 (the method overview diagram) is never referenced in the text; (2) the Observation section (Sec. 3) has abrupt transitions and introduces ARR/SD oracle without discussing their limitations; (3) the connection between the layer-wise analysis (Fig. 4) and the final hybrid design is not formally justified.

---

> ### Author Rebuttal · Authors · 2026-03-31
>
> We thank the reviewer for the careful and constructive feedback. We are encouraged that the reviewer finds the problem important and the empirical results strong. Below we clarify the main concerns and will revise the paper accordingly.
>
> ## ARR as a Practical Proxy, Not a Lossless Guarantee
>
> We agree that Dustin is **not** lossless, as verification uses a compressed KV cache. Our goal is to study the accuracy–efficiency trade-off under constrained budgets, rather than preserve the lossless guarantee of standard SD.
>
> We also clarify that the “SD oracle” is only an **upper bound under the ARR objective**, not an oracle for minimizing output divergence.
>
> While ARR is not a formal fidelity guarantee, it serves as a practical proxy for sparse selection quality. Empirically, we observe a strong negative correlation between ARR and KL divergence to full-KV logits at KV budget 512 (e.g., **-0.7486** on MultiNews and **-0.9625** on GovReport), indicating that higher ARR consistently corresponds to lower output distortion. We will clarify this in the revision.
>
>
> ## Non-Monotonic Alignment in Fig. 3
>
> We indeed observe that the 0.5B draft aligns better with 7B and 72B than with 14B and 32B. We agree this is interesting, but our main point is that this inconsistency is precisely why **draft-lookahead alone is insufficient**.
>
> Our layer-wise analysis in Sec. 3.4 / Fig. 4 shows that even when draft-lookahead is globally weaker, it remains useful in earlier layers, while target-history is more reliable in deeper layers. This complementarity is exactly the motivation for Dustin’s hybrid design. In other words, the non-monotonicity in Fig. 3 is not a contradiction to Dustin, but part of the empirical motivation for it.
>
> For the question on other draft scales, we agree broader validation would strengthen the claim, and we will clarify the current experimental scope more explicitly.
>
> ## Offline Search Overhead Is Small and Stable
>
> We agree that the practical cost of Algorithm 1 should be quantified more clearly.
>
> Since Reviewer YsWY raised the same concern, we provide the full wall-clock discussion in our response to Reviewer YsWY under **“Porting Overhead to New Models”** and summarize the key point here. In brief, the search introduces only a **small one-time offline cost** and does not require per-task recalibration.
>
> We also agree that the validation-set setup was not described clearly enough in the submission. The search is conducted using **50 LongReward samples**, and we will add this information to the paper. We further evaluated validation sizes of **25 / 50 / 100** and found that the resulting ARR on a held-out set differs by **less than 1%** across these settings, suggesting that the search is not highly sensitive to moderate changes in validation-set size.
>
> ## Failure Modes and Tail Cases
>
> We agree that averaged LongBench scores do not fully capture failure modes. Since this substantially overlaps with another reviewer’s question, we respectfully refer the reviewer to to our response to response on YsWY on **“Failure Modes and Distributional Quality Loss,”** where we provide tail-sensitive statistics and representative failure cases.
>
> The short conclusion is that degradation is **concentrated in a small subset of difficult examples**, especially tasks requiring precise long-range retrieval or exact answer spans, while moderate budgets such as 512 remain generally stable.
>
>
> ## Missing Baselines: SpecAttn and LongSpec
>
> We thank the reviewer for pointing out the missing comparisons and agree that this should be clarified.
>
> For **SpecAttn**, since no official implementation is publicly available, we implemented it ourselves and report the measured speedups below (target model: Q2.5-72B, batch size = 16):
>
> | Method   | 8K   | 16K  | 32K  |
> |---|---|---|---|
> | SpecAttn | 3.91× | 5.62× | 7.11× |
> | Dustin   | 4.76× | 7.06× | 9.17× |
>
> Dustin consistently outperforms SpecAttn across all context lengths.
>
> For **LongSpec**, its released implementation does not support **multi-batch or multi-GPU serving**, and additionally requires **training an auxiliary draft model**, making it difficult to obtain directly comparable throughput measurements under our evaluation setting.
>
> We also note that LongSpec and Dustin optimize **different bottlenecks**: LongSpec compresses the *draft model KV cache*, while Dustin targets the *target model KV cache*. These approaches are therefore complementary, and could potentially be combined. We will clarify this distinction and discussion in the revised version.
>
> ## Presentation and Clarity
>
> We thank the reviewer for the helpful feedback. We will revise the manuscript to improve clarity by (1) explicitly referencing Fig. 5 in the main text, (2) smoothing the transitions in Sec. 3 and clarifying the limitations of ARR and the SD oracle, and (3) making the connection between Fig. 4 and the hybrid design more explicit.

---

> > ### Author Rebuttal · Reviewer_tC4w · 2026-04-02
> >
> > The rebuttal substantially strengthens the paper, and I therefore upgrade my rating from `3 (Weak Reject)` to `4 (Weak Accept)`: it clarifies that ARR is a proxy (not lossless), provides strong ARR–KLD evidence, adds tail-sensitive failure analysis, explains the Fig. 3 non-monotonic alignment via layer-wise complementarity, quantifies offline/porting overhead, extends context/batch scaling, and supplies the missing SpecAttn baseline. Remaining gaps are mainly that the LongSpec comparison is not fully apples-to-apples under the same serving constraints, and further draft-scale/model-generalization validation would further increase confidence.
> >
> > Further questions:
> > In the final revision, could you provide an end-to-end comparison with LongSpec under the same multi-batch/multi-GPU serving constraints (or a clear, verifiable justification if direct comparison is impractical) and add additional validation across more draft scales/model families to further support the Fig. 3 non-monotonicity explanation and generalization?

---

> > > ### Author Response · Authors · 2026-04-07
> > >
> > > We thank the reviewer for the positive reassessment and for the helpful suggestions.
> > >
> > > **LongSpec comparison.** We agree that an end-to-end comparison under the same multi-batch / multi-GPU serving setting would strengthen the evaluation. We will include either (i) a controlled comparison under aligned serving constraints, or (ii) a more explicit and verifiable justification of any remaining discrepancies (e.g., training requirements and system limitations) in the camera-ready version.
> > >
> > > **Generalization across draft scales.** We also agree that broader validation would further support the explanation of the non-monotonic behavior in Fig. 3. We will extend our experiments to additional draft scales and model families to better characterize the generality of the hybrid strategy.**Generalization across draft scales.** For the non-monotonicity in Fig. 3, we agree that additional validation across a broader range of draft scales and model families would further strengthen the generalization claim, and we will include such results to better characterize the robustness of the hybrid strategy.

---

### Official Review · Reviewer_575q · 2026-03-10

**Soundness:** 3
**Presentation:** 4
**Significance:** 4
**Originality:** 3
**Overall Recommendation:** 3
**Confidence:** 4

**Summary:**

This paper introduces Dustin, a sparse verification framework designed to accelerate speculative decoding for long-context Large Language Models (LLMs).  In speculative decoding, the verification phase is often bottlenecked by the heavy memory-bandwidth costs associated with loading the Key-Value (KV) cache. Existing KV cache compression methods struggle in this regime: they either suffer from accuracy degradation due to permanent token eviction (saliency shift) or they introduce prohibitive computational overhead during dynamic selection.To overcome this, Dustin uses a hybrid attention aggregation strategy that fuses "lookahead" attention scores from the draft model with "historical" attention scores from the target model. The authors observed that draft-lookahead signals are highly accurate for early layers, while target-historical signals are more reliable for deeper layers. To prevent the importance estimation itself from becoming a latency bottleneck, Dustin utilizes Semantic Retrieval Heads (SRHs), performing importance scoring on only a minimal subset of attention heads. Evaluated on Qwen2.5 and Llama-3 models with up to 32k context lengths, Dustin achieves up to a 27.85x speedup in self-attention and a 9.17x end-to-end decoding speedup with negligible accuracy loss.

**Compliance With Llm Reviewing Policy:**

Affirmed.

**Final Justification:**

The rebuttal and additional sensitivity analyses have addressed the technical concerns regarding the robustness of the hybrid aggregation strategy and the stability of Semantic Retrieval Heads across varying context lengths. The supplementary evidence supports the technical soundness and the reported speedups in long-context speculative decoding.

**Key Questions For Authors:**

The budget allocation parameter controlling the ratio of draft-lookahead to target-historical tokens is fixed after offline optimization. How sensitive is the framework if deployed on highly out-of-distribution tasks where the draft model's lookahead is completely misaligned?

The current framework selects from the full history and does not permanently evict tokens to reduce the actual vram footprint. Are there fundamental barriers to combining Dustin's sparse verification with actual cache eviction methods to simultaneously improve throughput and expand the maximum supportable context length?

The overhead of estimating token importance is kept low by using semantic retrieval heads. How robust is the offline profiling of these specific heads when shifting between wildly different context lengths or prompt structures?

**Limitations:**

Yes, the authors adequately discuss limitations in Appendix H. They explicitly acknowledge that the static budget allocation parameter leaves a performance gap compared to a dynamic allocation strategy. Furthermore, they highlight that Dustin does not inherently expand the maximum supportable context length constrained by GPU memory capacity, as it uses a global indexing mechanism and does not physically remove tokens to reduce the vram footprint.

**Strengths And Weaknesses:**

Soundness: The empirical evaluation demonstrates substantial speedups across varying context lengths (8k to 32k) and batch sizes (8 and 16). The ablation studies effectively isolate the contributions of the target-historical and draft-lookahead signals. These ablations prove the necessity of the hybrid approach to maintain accuracy across diverse benchmarks. A minor weakness is that the budget allocation parameter balancing draft and target signals is statically tuned offline. Different tasks exhibit distinct preferences for validation signals, meaning a static scalar leaves a performance gap compared to an ideal dynamic strategy.

Presentation: The paper clearly articulates the specific bottlenecks in long-context speculative decoding. Visualizations, such as the layer-wise attention recovery analysis, perfectly motivate the core hybrid design choice. The writing is logically structured, moving smoothly from empirical observations of signal decay to the proposed architectural solution.

Significance: Scaling llms to long contexts while maintaining high throughput is a pressing challenge in deployment. Achieving a 9.17x end-to-end decode-stage throughput improvement on a 72b parameter model at a 32k context length is a highly significant practical result.

Originality: Combining the draft model's lookahead foresight with the target model's historical reliability to predict token importance is a novel synthesis. Applying semantic retrieval heads to minimize the overhead of online estimation within the speculative decoding pipeline is a clever architectural adaptation.

---

> ### Author Rebuttal · Authors · 2026-03-31
>
> We thank the reviewer for the insightful questions and constructive feedback.
>
> ## Robustness under Draft–Target Misalignment (OOD Sensitivity)
>
> This failure mode is already reflected in Fig. 3: the 0.5B draft model exhibits noticeably weaker alignment for Qwen2.5-14B and Qwen2.5-32B, where the ARR gap to the oracle is significantly larger than for 7B and 72B. This setting can be viewed as a proxy for out-of-distribution (OOD) conditions where the draft lookahead becomes unreliable.
>
> Dustin is explicitly designed to mitigate such misalignment through its **hybrid aggregation**. Rather than relying solely on draft-lookahead, it combines draft-lookahead with target-history signals, leveraging their layer-wise complementarity:
> - draft-lookahead provides strong query-aware signals in early layers,
> - target-history provides stable and model-consistent signals in deeper layers.
>
> As a result, when draft lookahead degrades, the target-history branch naturally compensates.
>
> Empirically, even in these more challenging 14B/32B settings, Dustin maintains accuracy close to FullKV. For example:
> - **Qwen2.5-32B**: 46.85 / 45.72 / 43.28 (KV=512/256/128) vs. 47.52 (FullKV)
> - **Qwen2.5-14B**: 51.46 / 50.31 / 48.06 vs. 52.27 (FullKV)
>
> These results indicate that the framework degrades gracefully even under substantial draft–target misalignment.
>
> **Sensitivity to budget allocation.**
> Under severe misalignment, allocating more budget to the target-history branch consistently improves robustness, as reflected in our ablations. In practice, we observe that performance remains stable across a relatively broad range of allocation ratios, suggesting that the gap to an optimal dynamic policy is limited. Extending Dustin with adaptive allocation is a natural direction and can be incorporated without modifying the core framework.
>
> ---
>
> ## Compatibility with KV Cache Eviction and Fundamental Trade-offs
>
> Dustin and KV eviction methods operate on **orthogonal design axes**:
>
> - **Eviction methods** reduce *persistent memory footprint* by permanently removing tokens.
> - **Dustin** reduces *verification-time latency* by sparsifying KV access during attention.
>
> There is no fundamental barrier to combining the two. A hybrid system could naturally integrate **conservative eviction** to control memory footprint, while leveraging Dustin to minimize verification-time bandwidth, addressing both capacity and latency simultaneously.
>
> The key distinction lies in **reversibility**.
> Eviction is inherently irreversible and thus sensitive to **saliency shift**, where tokens that appear unimportant at one step may become important later. Once evicted, such tokens cannot be recovered.
>
> In contrast, Dustin performs **query-dependent, step-wise selection over the full KV cache**, allowing it to adapt to dynamically changing token importance without permanent information loss.
>
> Importantly, this difference does not make the two approaches incompatible, but instead highlights a fundamental trade-off between **memory capacity (eviction)** and **selection flexibility (Dustin)**.
>
> Our work focuses on the dominant bottleneck in long-context speculative decoding—**memory bandwidth during verification**—under a fixed KV budget. Extending Dustin with eviction-based memory reduction is a promising direction, but would require a different evaluation setting (e.g., scaling maximum context length under fixed VRAM), which we leave for future work.
>
> ---
>
> ## Robustness of Semantic Retrieval Heads (SRHs) under Distribution Shift
>
> Our approach follows prior work (Lin et al., 2025), where semantic retrieval heads (SRHs) are identified once per model and then fixed without task-specific recalibration.
>
> To assess robustness, we conduct sensitivity analysis along three axes:
> 1. calibration size (subsampling),
> 2. prompt/data variation (different haystack sources),
> 3. context length scaling.
>
> Across all settings, SRHs remain highly stable:
>
> - **Context length shift (16K/32K → 50K)**:
>   Jaccard overlap ≥ 0.88, Spearman’s ρ ≈ 0.99
> - **Prompt/data variation**:
>   overlap ≥ 0.94, ρ ≈ 0.99
>
> These results indicate that SRHs primarily capture **model-internal retrieval behavior**, rather than overfitting to specific calibration data or prompt distributions.
>
> Consequently, offline profiling generalizes well across diverse settings while maintaining negligible overhead, eliminating the need for per-task re-profiling.

---

> > ### Author Rebuttal · Reviewer_575q · 2026-04-03
> >
> > My concerns have been adequately addressed. Thanks for the detailed rebuttal.

---

> > > ### Author Response · Authors · 2026-04-08
> > >
> > > We thank the reviewer for the positive acknowledgement and thoughtful final assessment.
> > >
> > > We are glad that the additional analyses helped address the technical concerns regarding the robustness of the hybrid aggregation strategy and the stability of SRH selection. We will incorporate the relevant clarifications and supporting results into the final version.
> > >
> > > We appreciate the reviewer’s constructive feedback throughout the discussion.

---

### Official Review · Reviewer_YsWY · 2026-03-10

**Soundness:** 3
**Presentation:** 3
**Significance:** 3
**Originality:** 3
**Overall Recommendation:** 4
**Confidence:** 4

**Summary:**

The paper proposes Dustin, a sparse verification framework for speculative decoding in long‑context, multi‑batch LLM inference. It observes that verification is dominated by KV‑cache loading at long contexts and large batch sizes. Dustin fuses target‑historical attention and draft‑lookahead attention to select a small set of critical KV entries, and uses semantic retrieval heads (SRHs) to estimate importance efficiently. On PG‑19 and LongBench with Qwen2.5‑72B, the method reports large self‑attention speedups (up to 27.85×) and end‑to‑end decode‑stage throughput gains (up to 9.17×) with minimal accuracy loss.

**Compliance With Llm Reviewing Policy:**

Affirmed.

**Final Justification:**

Thank you for the detailed rebuttal. However, my concerns are partially resolved. I will keep my score.

**Key Questions For Authors:**

- How robust are the gains in settings with shorter contexts (e.g., 4–8k) or smaller batch sizes?
- Can you provide a more systematic analysis of quality loss (e.g., distributional metrics or failure modes) beyond a few benchmarks?
- How sensitive are results to SRH selection and offline profiling choices, and what is the overhead to port them to a new model?

**Limitations:**

yes

**Strengths And Weaknesses:**

Strengths
- Targets a real bottleneck in long‑context speculative decoding: KV‑cache access during verification.
- The hybrid signal design (history + lookahead) is well motivated by ARR analysis and supported by ablations.
- The SRH‑based estimator makes the online importance computation practical, avoiding full attention reconstruction.
- Empirical gains are substantial in the intended regime (long context, large batch), and scaling trends are consistent.

Weaknesses
- The reported 9× decode‑stage speedup is compelling but largely tied to a specific regime (32k context, batch 16, heavy verification dominance). It is unclear how much benefit remains in shorter contexts or smaller batch sizes typical in many deployments.
- The method is inherently lossy due to KV compression. While reported accuracy drops are small, there is no theoretical bound or broader distributional analysis of output deviation.
- The approach relies on offline profiling and SRH selection; the portability of these choices across models, tasks, and system stacks is not fully explored.
- The evaluation focuses on a limited set of model families (Qwen2.5/Llama3). Broader evidence on other architectures or instruction‑tuned settings is missing.

---

> ### Author Rebuttal · Authors · 2026-03-31
>
> We thank the reviewer for the thoughtful and constructive feedback. Below we address each concern in detail.
>
> ---
> ## Scaling Behavior Across Context Lengths and Batch Sizes
>
> To assess Dustin beyond the 32K / batch-16 regime, we additionally sweep context length and batch size.
>
> Dustin already provides meaningful speedups at shorter contexts and smaller batch sizes (1.50×–3.57×), and the gain increases consistently with both context length and batch size, reaching 9.17× at 32K / batch-16.
>
> | Batch Size | 4K | 8K | 16K | 32K |
> |------------|------|------|------|------|
> | 2  | 1.50x | 1.60x | 1.74x | 2.19x |
> | 4   | 1.98x | 2.19x | 2.84x | 3.99x |
> | 8   | 2.47x | 3.01x | 4.45x | 6.61x |
> | 16  | 3.57x | 4.76x | 7.06x | 9.17x |
>
> This trend is expected: Dustin targets KV-cache-loading-dominated verification, so its relative benefit naturally grows as context length and batch size increase.
>
> ---
>
> ## Failure Modes and Distributional Quality Loss
>
> We agree that aggregate metrics do not fully capture quality degradation. We therefore analyze the three worst-performing LongBench tasks using tail-sensitive statistics: **Mean Drop** (average score drop from the full-cache baseline), **P90/P95 Drop** (90th/95th percentile of per-example score drop), and **No-regression Rate** (fraction of examples whose score is unchanged or higher).
>
> | Task | KV Budget | Mean Drop | P90 Drop | P95 Drop | No-regression Rate |
> |---|---:|---:|---:|---:|---:|
> | passage_retrieval_en | 128 | 0.1575 | 1.0000 | 1.0000 | 84.0% |
> | passage_retrieval_en | 512 | 0.0000 | 0.0000 | 0.0000 | 100.0% |
> | samsum | 128 | 0.0698 | 0.2178 | 0.3552 | 29.0% |
> | samsum | 512 | 0.0226 | 0.1491 | 0.2473 | 45.0% |
> | musique | 128 | 0.0284 | 0.0313 | 0.7570 | 88.5% |
> | musique | 512 | 0.0110 | 0.0000 | 0.3511 | 93.0% |
>
> The degradation is task-dependent and concentrated in tail cases. In particular:
> - **passage_retrieval_en**: near-miss retrieval errors (selecting a neighboring passage),
> - **samsum**: boundary/format drift rather than semantic corruption,
> - **musique**: occasional factual substitution or omission of key answer spans.
>
> We also observe representative examples such as predicting a **neighboring but incorrect passage**, or producing a **partially correct answer with missing key entities** (e.g., shortening *“Kenton County, Kentucky”* to *“Kentucky”*). These errors are structured rather than arbitrary.
>
> These results suggest that sparse KV approximation is generally safe under moderate budgets (e.g., 512), while tighter budgets (e.g., 128) mainly impact tasks requiring **precise long-range retrieval or exact answer spans**.
>
> Overall, the quality loss is concentrated in a small subset of difficult examples rather than broadly distributed. We will include these tail statistics and representative cases in the final version.
>
> ---
>
> ## Robustness and Portability of SRH Selection
>
> ### Sensitivity to Profiling Choices
>
> Our SRH identification strictly follows Lin et al. (2025), where SRHs are identified **once per model** and then kept fixed.
>
> Their results show that SRHs transfer across tasks without recalibration, indicating that they capture **model-internal retrieval behavior** rather than dataset-specific artifacts.
>
> We further evaluate SRH stability under:
> 1. calibration set size
> 2. haystack source
> 3. context length
>
> Using top-64 SRHs, we observe:
>
> - **Jaccard overlap: 0.88–1.00**
> - **Spearman’s ρ: 0.99 (consistent across all settings)**
>
> | Setting | Top-k | Jaccard overlap ↑ | Spearman’s ρ ↑ |
> |---|---:|---:|---:|
> | 0.25 vs. 1.0 calibration size | 64 | 0.94 | 0.99 |
> | 0.50 vs. 1.0 calibration size | 64 | 1.00 | 0.99 |
> | part1 vs. part2 | 64 | 1.00 | 0.99 |
> | part1 vs. part3 | 64 | 0.94 | 0.99 |
> | 16K vs. 50K | 64 | 0.88 | 0.99 |
> | 32K vs. 50K | 64 | 0.94 | 0.99 |
>
> This indicates that SRH selection is **highly stable** and not sensitive to moderate profiling changes.
>
> ---
>
> ### Porting Overhead to New Models
>
> SRH identification is adopted from Lin et al. (2025). We re-measured their pipeline on Llama-3.1-8B-Instruct on A100:
>
> | Max input length | Identification time |
> |---:|---:|
> | 16K | 13 min |
> | 32K | 19 min |
> | 50K | 27 min |
>
> After SRHs are identified, Dustin incurs a **one-time offline cost** consisting of:
> - a full-cache run for attention collection
> - Algorithm 1 search
>
> **Measured overhead:**
>
> **H100 ×1**
> - L3.1 8B + 1B: 7.5 min (collection) + 4.4 min (search) ≈ **11.9 min total**
> - Q2.5 7B + 0.5B: 6.6 min + 7.1 min ≈ **13.8 min total**
> - Q2.5 14B + 0.5B: 10.6 min + 7.5 min ≈ **18.2 min total**
>
> **H100 ×2**
> - Q2.5 32B + 0.5B: 15.8 min + 8.3 min ≈ **24.1 min total**
> - Q2.5 72B + 0.5B: 26.6 min + 8.8 min ≈ **35.4 min total**
>
> Overall, Dustin requires **no per-task recalibration** and only a **small one-time cost**.
>
> ---
>
> ## Generality Across Model Families
> We agree that our evaluation focuses on Qwen2.5 and Llama3, which are representative open-weight LLMs.
>
> We will include additional model families in the camera-ready version.

---

> > ### Author Rebuttal · Reviewer_YsWY · 2026-04-03
> >
> > Thank you for the detailed response. We appreciate the additional experiments and analyses, which have largely addressed our concerns.

---

> > > ### Author Response · Authors · 2026-04-08
> > >
> > > We thank the reviewer for the positive follow-up and are encouraged that the additional experiments and analyses have largely addressed the concerns.
> > >
> > > We will incorporate the relevant clarifications and supporting results into the final version, and further improve the discussion of generalization and practical deployment considerations.
> > >
> > > We appreciate the reviewer’s careful reading and constructive feedback.

---

### Official Review · Reviewer_jMVv · 2026-03-13

**Soundness:** 3
**Presentation:** 4
**Significance:** 3
**Originality:** 3
**Overall Recommendation:** 5
**Confidence:** 3

**Summary:**

The paper proposes an sparse verification method in long-context speculative decoding where the tokens for verification is selected efficiently. The paper conducts an analysis that either target model attention (based on the latest token) or draft model attention (based on draft tokens) solely not sufficient to select the tokens and proposes a hybrid policy from 1) target model history attention and 2) draft model lookahead attention using only Semantic Retrieval Heads. In evaluation, it shows the proposed method can achieve up to 9.17x at 32k context length with Llama3 and Qwen2.5 models on LongBench and PG-19 outperforming sparse attention baselines in accuracy.

**Compliance With Llm Reviewing Policy:**

Affirmed.

**Final Justification:**

Most of my concerns are addressed from the rebuttal except that the competing methods in the comparison study might not be in their best settings, which is not 100% clear to me. Nevertheless, it seems a good paper with clear presentation, that tries to fuse two different problems (speculative decoding and long context) and demonstrate an efficient approach. I raise my recommendation to 5.

**Key Questions For Authors:**

Q1. Could authors give more detailed discussion about comparison to Quest to ensure fairness in test setting? Quest seems to be chosen as sparse attention baseline which operates in page level while the proposed method works in token level, which may result in unfair comparison. Test setting may be favorable to the proposed method over Quest. Due to page-wise nature of Quest, with very small token budget, its token coverage can be limited compared to token-wise methods.

Q2. Could Quest also be included in efficiency comparison analysis?

Q2. Could authors give more details about speculative decoding setting such as draft model size, draft token structure, and draft length? This will be helpful to understand costs for drafting and verification.

**Limitations:**

yes

**Strengths And Weaknesses:**

**Strengths**
- Significant reduction of computation overheads for KV selection in sparse attention by leveraging draft model
- Significant acceleration of attention computation in draft verification and end-to-end throughput
- Optimization-based Semantic Retrieval Head search for attention score computation overhead reduction based on the attention recovery rate

**Weaknesses**
- Presents ARR but how it is related to the significance of selected (recoved) tokens is not clear, i.e., error bound analysis in attention outputs
- Some lack of details in test settings such as draft model sizes, draft lengths, hyperparameters for baselines. Such information can be important since draft model needs full context attention that cause large overhead in drafting depending on the configuration.
- StremingLLM is chosen as a cache eviction baseline which is considered to be least powerful. More advanced with better accuracy yet efficient cache eviction method could have been chosen for comparison

---

> ### Author Rebuttal · Authors · 2026-03-30
>
> We appreciate the reviewer’s positive feedback and constructive suggestions. Below we address the key questions and concerns.
>
> ---
> ## Fairness of Comparison with Quest (Token-level vs Page-level)
> To ensure fairness in the accuracy comparison, we follow the official Quest setup without modifying its recommended configuration. In particular, we use **page_size = 16**, which is the official setting adopted in Quest.
>
> Quest’s page-level design is fundamental to its efficiency rather than an arbitrary implementation choice. It partitions the KV cache into pages and summarizes each page using **min/max key statistics**, resulting in a scoring cost of:
>
> - **O(2 × (N / P))**, where *P* is the page size.
>
> When reducing the page size:
> - At **P = 2**, the cost already becomes **O(N)**, comparable to full token-wise attention.
> - At **P = 1 (token-level)**, the cost becomes **O(2N)**, i.e., roughly **2× full-cache QK computation**.
>
> Thus, Quest cannot be made token-level without losing its efficiency advantage, as its design inherently trades granularity for computational savings. In fact, if Quest were configured at the token level, its forward latency would exceed that of full-cache attention, making it an impractical and unfavorable setting; therefore, we do not include such a configuration in our evaluation.
>
> In contrast, Dustin reduces cost along a different axis by restricting score computation to a small subset of **Semantic Retrieval Heads (SRHs)**. This enables **token-level importance estimation** with negligible overhead (~1% of full attention at long context), as shown in Fig. 8.
>
> ---
>
> ## Efficiency Comparison with Quest
>
> Although Quest is not explicitly included in the end-to-end throughput table, its efficiency can be inferred from two factors:
>
> **(1) Higher lower-bound latency in verification.**
> Quest leaves layers 0–1 uncompressed, meaning part of the KV cache is always accessed in full. Dustin applies token-level selection across all layers, resulting in consistently reduced KV usage. This leads to a lower verification latency floor for Dustin.
>
> **(2) Higher importance estimation overhead.**
> Fig. 8 shows that Quest-16 incurs significantly higher normalized latency, while Dustin remains lightweight (~1% of full attention). This gap increases with sequence length, reflecting the higher cost of page-level scoring.
>
> Combining these factors, Quest is expected to be consistently slower than Dustin in an end-to-end speculative decoding setting under the same KV budget.
>
> ---
>
> ## Speculative Decoding Configuration Details
>
> Key configuration details are as follows:
>
> - **Draft model size:** lightweight same-family models (e.g., Qwen2.5-72B paired with Qwen2.5-0.5B), as described in Sec. 5.1
> - **Speculative depth:** 4 for 72B/32B models, and 3 for smaller models (Appendix B.4)
> - **Draft structure:** standard **sequential draft blocks** (not tree-based)
>
> These details will be explicitly clarified in the final version for completeness.
>
> ---
>
> ## ARR vs Output Error (Attention–Output Relationship)
>
> We validate ARR's relationship to output distortion through an additional experiment:
>
> - Fix KV budget = 512
> - Run full-cache forward pass and treat its output logits as the reference
> - Randomly sample KV subsets of size 512
> - For each subset, run sparse forward pass and compute:
>   (1) ARR, and (2) KL divergence between sparse logits and full-cache logits
>
> Random sampling removes bias from specific selection policies, allowing us to isolate the intrinsic relationship between retained attention mass (ARR) and output distortion (KL divergence).
>
> **Results show strong negative correlation:**
>
> | Dataset   | ARR–KLD Correlation |
> |---|---|
> | MultiNews | -0.75  |
> | GovReport | -0.96 |
>
> We also observe a trend where higher ARR consistently leads to lower KL divergence.
>
> These results suggest that ARR is a strong indicator of output fidelity: higher ARR is consistently associated with lower output distortion. To further support this observation, we will include the corresponding ARR–KLD plots in the appendix of the final version.
>
> ---
>
> ## Stronger Baselines Beyond StreamingLLM
>
> Following the suggestion, we additionally evaluate **SnapKV** as a stronger baseline. The results show that Dustin maintains strong accuracy performance. These results and discussion will be included in the revised version.
>
> | Model | Method | KV Budget | Single-doc QA | Multi-doc QA | Summarization | Few-shot | Synthetic | Code | Avg. |
> |---|---|---:|---:|---:|---:|---:|---:|---:|---:|
> | Q2.5-72B | Vanilla | - | 44.23 | 57.00 | 27.54 | 72.55 | 60.00 | 66.70 |55.81|
> |  | SnapKV | 512 | 42.80 | 58.77 | 23.02 | 71.21 | 55.00 | 57.29 |51.90|
> |  | Dustin | 512 | 44.23 | 56.35 | 27.10 | 71.23 | 60.00 | 65.97 |55.23|
> | L3.3-70B | Vanilla | - | 45.48 | 58.88 | 28.58 | 70.75 | 53.63 | 49.64 |50.94|
> |  | SnapKV | 512 | 44.45 | 59.58 | 24.87 | 68.75 | 53.39 | 58.82 |52.41|
> |  | Dustin | 512 | 45.27 | 59.10 | 27.76 | 70.14 | 53.40 | 59.15 |53.23|

---

> > ### Author Rebuttal · Reviewer_jMVv · 2026-04-03
> >
> > Thank you for the response. The rebuttal provides helpful clarification for most concerns. However, since efficiency and accuracy are inherently coupled in selective attention schemes, it would be important to use consistent baselines for comparison. In particular, Quest may benefit from paged attention in terms of end‑to‑end latency, potentially allowing a larger effective KV budget at the same latency compared to token‑wise selection. To clarify this trade‑off, I suggest either including Quest in the throughput (or end‑to‑end latency) comparison or providing a Pareto efficiency plot.

---

> > > ### Author Response · Authors · 2026-04-08
> > >
> > > We thank the reviewer for the suggestion. To further improve fairness, we will include Quest with multiple page sizes in the accuracy evaluation in the revised version.
> > >
> > > We agree that the key comparison should be in terms of the accuracy–latency trade-off. To this end, we attempted to include Quest in the end-to-end throughput evaluation. However, the publicly available implementation has important limitations for our setting (e.g., no optimized GQA support and no multi-batch support).
> > >
> > > To better approximate this regime, we implemented a FlashInfer-based version of Quest. While this implementation may not fully reflect an optimized Quest kernel, it allows us to estimate its practical behavior under the same setting. The measured end-to-end throughput speedup (over full-cache forward) on Llama 3 8B with batch size 8 on H200 is:
> > >
> > > - 8K: Quest 1.15× vs Dustin 1.85×
> > > - 16K: Quest 1.23× vs Dustin 2.40×
> > > - 32K: Quest 1.33× vs Dustin 2.85×
> > >
> > > These results suggest that even when accounting for page-wise efficiency, Dustin achieves a more favorable accuracy–efficiency trade-off, due to its lightweight SRH-based scoring and full-layer KV reduction. We will include these results and a corresponding discussion in the revised version.

---

### Decision · Program_Chairs · 2026-04-30

**Decision:**

Accept (regular)

**Comment:**

This paper proposes an augmented sparse verification strategy for efficient long-context speculative decoding, combining draft-model lookahead with target-model historical signals. Across reviews, there is consensus that the work is well-motivated, clearly presented, and practically impactful. Reviewers highlight strong empirical results, particularly substantial throughput gains with minimal quality degradation in long-context settings, and view the method as a meaningful systems contribution.

The primary concerns focus on three areas: (1) limited theoretical justification for ARR as a proxy for fidelity, (2) evaluation gaps—especially around broader baselines, robustness outside the long-context regime, and fairness of comparisons, and (3) insufficient clarity on overheads (e.g., offline profiling/search costs) and failure cases.

The rebuttal addressed many of these issues by adding empirical validation (e.g., ARR-KL alignment), clarifying overhead trade-offs, and expanding discussion of limitations and comparisons. This led to improved reviewer confidence, with at least one score increase and others noting that key concerns were largely resolved.

Overall, the paper is viewed positively. It offers a solid and impactful contribution with strong practical relevance, though some residual concerns remain regarding theoretical grounding and completeness of evaluation.